# Recursive PAC-Bayes: A Frequentist Approach to Sequential Prior Updates with No Information Loss

**Yi-Shan Wu**
University of South Denmark
yswu@imada.sdu.dk

**Yijie Zhang**
University of Copenhagen & Novo Nordisk A/S
yizh@di.ku.dk

**Badr-Eddine Chérief-Abdellatif**
CNRS, LPSM, Sorbonne Université, Université Paris Cité
badr-eddine.cherief-abdellatif@cnrs.fr

**Yevgeny Seldin**
University of Copenhagen
seldin@di.ku.dk

## Abstract

PAC-Bayesian analysis is a frequentist framework for incorporating prior knowledge into learning. It was inspired by Bayesian learning, which allows sequential data processing and naturally turns posteriors from one processing step into priors for the next. However, despite two and a half decades of research, the ability to update priors sequentially without losing confidence information along the way remained elusive for PAC-Bayes. While PAC-Bayes allows construction of data-informed priors, the final confidence intervals depend only on the number of points that were not used for the construction of the prior, whereas confidence information in the prior, which is related to the number of points used to construct the prior, is lost. This limits the possibility and benefit of sequential prior updates, because the final bounds depend only on the size of the final batch.

We present a novel and, in retrospect, surprisingly simple and powerful PAC-Bayesian procedure that allows sequential prior updates with no information loss. The procedure is based on a novel decomposition of the expected loss of randomized classifiers. The decomposition rewrites the loss of the posterior as an excess loss relative to a downscaled loss of the prior plus the downscaled loss of the prior, which is bounded recursively. As a side result, we also present a generalization of the split-kl and PAC-Bayes-split-kl inequalities to discrete random variables, which we use for bounding the excess losses, and which can be of independent interest. In empirical evaluation the new procedure significantly outperforms state-of-the-art.

## 1 Introduction

PAC-Bayesian analysis was born from an attempt to derive frequentist generalization guarantees for Bayesian-style prediction rules (Shawe-Taylor and Williamson, 1997, McAllester, 1998). The motivation was to provide a way to incorporate prior knowledge into the frequentist analysis of generalization. PAC-Bayesian bounds provide high-probability generalization guarantees for randomized classifiers. A randomized classifier is defined by a distribution $\rho$ on a set of prediction rules $\mathcal{H}$, which is used to sample a prediction rule each time a prediction is to be made. Bayesian posterior is an example of a randomized classifier, whereas PAC-Bayesian bounds hold generally for all randomized classifiers. Prior knowledge is encoded through a prior distribution $\pi$ on $\mathcal{H}$, and the complexity of a posterior distribution $\rho$ is measured by the Kullback-Leibler (KL) divergence from the prior, $\text{KL}(\rho\|\pi)$. PAC-Bayesian generalization guarantees are optimized by posterior distributions $\rho$ that optimize a trade-off between empirical data fit and divergence from the prior in the KL sense.

38th Conference on Neural Information Processing Systems (NeurIPS 2024).

Selection of a "good" prior plays an important role in the PAC-Bayesian bounds. If one manages to foresee which prediction rules are likely to produce low prediction error and allocate a higher prior mass for them, then the bounds are tighter, because the posterior only needs to make a small deviation from the prior. But if the prior mass on well-performing prediction rules is small, the bounds are loose. A major technique to design good priors is to use part of the data to estimate a good prior and the rest of the data to compute a PAC-Bayes bound. It is known as data-dependent or data-informed priors (Ambroladze et al., 2007). However, all existing approaches to data-informed priors have three major disadvantages. The first is that the bounds are computed on "the rest of the data" that were not used in construction of the prior. Thus, the sample size in the bounds is only a fraction of the total sample size. Therefore, empirically data-informed priors are not always helpful. In many cases starting with an uninformed prior and using all the data to compute the posterior and the bound turns to be superior to sacrificing part of the data for prior construction (Ambroladze et al., 2007, Mhammedi et al., 2019). The second disadvantage is that all the confidence information about the prior is lost in the process. In particular, a prior trained on a few data points is treated in the same way as a prior trained on a lot of data. And a third related disadvantage is that sequential data processing provides no benefit, because the bounds only depend on the size of the last chunk and all the confidence information from processing earlier chunks is lost in the process.

Our main contribution is a new (and simple) way of decomposing the loss of a randomized classifier defined by the posterior. We write it as an excess loss relative to a downscaled loss of the randomized classifier defined by the prior plus the downscaled loss of the randomized classifier defined by the prior. The excess loss can be bounded using PAC-Bayes-Empirical-Bernstein-style inequalities (Tolstikhin and Seldin, 2013, Mhammedi et al., 2019, Wu et al., 2021, Wu and Seldin, 2022), whereas the loss of the randomized classifier defined by the prior can be bounded recursively. The recursive bound can both use the data used for construction of the prior and "the rest of the data", and thereby preserves confidence information on the prior. Our contribution stands out relative to all prior work on PAC-Bayes, and in fact all prior work on frequentist generalization bounds, because it makes sequential data processing and sequential prior updates meaningful and beneficial.

We note that while several recent papers experimented with sequential posterior updates by using martingale-style analysis, in all these works the prior remained fixed and only the posterior was changing (Chugg et al., 2023, Biggs and Guedj, 2023, Rodríguez-Gálvez et al., 2024). The work on sequential posterior updates is orthogonal to our contribution and can be combined with it. Namely, it is possible to apply sequential posterior updates in-between sequential prior updates. Another line of work used tools from online learning to derive PAC-Bayesian bounds (Jang et al., 2023), and in this context Haddouche and Guedj (2023) have used sequential prior updates, but their bounds hold for a uniform aggregation of sequentially constructed posteriors, which is different from standard posteriors studied in our work. The confidence bounds in their work come primarily from aggregation rather than confidence in individual posteriors in the sequence (the denominator of their bounds depends on the number of aggregated posteriors). The need to construct and maintain a large number of posteriors has a negative impact on the computational efficiency. Our work is the first one allowing sequential prior updates without loss of confidence information.

An additional side contribution of independent interest is a generalization of the split-kl and PAC-Bayes-split-kl inequalities of Wu and Seldin (2022) from ternary to general discrete random variables. It is based on a novel representation of discrete random variables as a superposition of Bernoulli random variables.

The paper is organized in the following way. In Section 2 we briefly survey the evolution of data-informed priors in PAC-Bayes and present our main idea behind Recursive PAC-Bayes; in Section 3 we present our generalization of the split-kl and PAC-Bayes-split-kl inequalities, which are later used to bound the excess losses; in Section 4 we present the Recursive PAC-Bayes bound; in Section 5 we present an empirical evaluation; and in Section 6 we conclude with a discussion.

## 2 The evolution of data-informed priors and the idea of Recursive PAC-Bayes

In this section we briefly survey the evolution of data-informed priors, and then present our construction of Recursive PAC-Bayes. We consider the standard classification setting, with $\mathcal{X}$ being a sample space, $\mathcal{Y}$ a label space, $\mathcal{H}$ a set of prediction rules $h : \mathcal{X} \to \mathcal{Y}$, and $\ell(h(X), Y) = \mathbb{1}(h(X) \neq Y)$ the zero-one loss function, where $\mathbb{1}(\cdot)$ denotes the indicator function. We let $\mathcal{D}$

denote a distribution on $\mathcal{X} \times \mathcal{Y}$ and $S = \{(X_1, Y_1), \ldots, (X_n, Y_n)\}$ an i.i.d. sample from $\mathcal{D}$. Let $L(h) = \mathbb{E}_{(X,Y) \sim \mathcal{D}}[\ell(h(X), Y)]$ be the expected and $\hat{L}(h, S) = \frac{1}{n} \sum_{i=1}^{n} \ell(h(X_i), Y_i)$ the empirical loss.

Let $\rho$ be a distribution on $\mathcal{H}$. A *randomized classifier* associated with $\rho$ samples a prediction rule $h$ according to $\rho$ for each sample $X \in \mathcal{X}$, and applies it to make a prediction $h(X)$. The expected loss of such randomized classifier, which we call $\rho$, is $\mathbb{E}_{h \sim \rho}[L(h)]$ and the empirical loss is $\mathbb{E}_{h \sim \rho}[\hat{L}(h, S)]$. For brevity we use $\mathbb{E}_{\rho}[\cdot]$ to denote $\mathbb{E}_{h \sim \rho}[\cdot]$.

We use $\mathrm{KL}(\rho \| \pi)$ to denote the Kullback-Leibler divergence between two probability distributions, $\rho$ and $\pi$ (Cover and Thomas, 2006). For $p, q \in [0, 1]$ we further use $\mathrm{kl}(p \| q) = \mathrm{KL}((1-p, p) \| (1-q, q))$ to denote the Kullback-Leibler divergence between two Bernoulli distributions with biases $p$ and $q$.

The goal of PAC-Bayes is to bound $\mathbb{E}_{\rho}[L(h)]$. Below we present how the bounds on $\mathbb{E}_{\rho}[L(h)]$ have evolved. In Appendix A we also provide a graphical illustration of the evolution.

**Uninformed priors**  Early work on PAC-Bayes used *uninformed priors* (McAllester, 1998). An uniformed prior $\pi$ is a distribution on $\mathcal{H}$ that is independent of the data $S$. A classical, and still one of the tightest bounds, is the following.

**Theorem 1** (PAC-Bayes-kl Inequality, Seeger, 2002, Maurer, 2004). *For any probability distribution $\pi$ on $\mathcal{H}$ that is independent of $S$ and any $\delta \in (0, 1)$:*

$$\mathbb{P}\left(\exists \rho \in \mathcal{P} : \mathrm{kl}\left(\mathbb{E}_{\rho}[\hat{L}(h, S)] \middle\| \mathbb{E}_{\rho}[L(h)]\right) \geq \frac{\mathrm{KL}(\rho \| \pi) + \ln(2\sqrt{n}/\delta)}{n}\right) \leq \delta,$$

*where $\mathcal{P}$ is the set of all probability distributions on $\mathcal{H}$, including those dependent on $S$.*

A posterior $\rho$ that minimizes $\mathbb{E}_{\rho}[L(h)]$ has to balance between allocating higher mass to prediction rules $h$ with small $\hat{L}(h, S)$ and staying close to $\pi$ in the $\mathrm{KL}(\rho \| \pi)$ sense. Since $\pi$ has to be independent of $S$, typical uninformed priors aim "to leave maximal options open" for $\rho$ by staying close to uniform.

**Data-informed priors**  Ambroladze et al. (2007) proposed to split the data $S$ into two disjoint sets, $S = S_1 \cup S_2$, and use $S_1$ to construct a *data-informed prior* $\pi$ and compute a bound on $\mathbb{E}_{\rho}[L(h)]$ using $\pi$ and $S_2$. Since in this approach $\pi$ is independent of $S_2$, Theorem 1 can be applied. The advantage is that $\pi$ can use $S_1$ to give higher mass to promising classifiers, thus relaxing the regularization pressure $\mathrm{KL}(\rho \| \pi)$ and making it easier for $\rho$ to allocate even higher mass to well-performing classifiers (those with small $\hat{L}(h, S_2)$). The disadvantage is that the sample size in the bound (the $n$ in the denominator) decreases from the size of $S$ to the size of $S_2$. Indeed, Ambroladze et al. observed that the sacrifice of $S_1$ for prior construction does not always pay off.

**Data-informed priors + excess loss**  Mhammedi et al. (2019) observed that if we have already sacrificed $S_1$ for the construction of $\pi$, we could also use it to construct a reference prediction rule $h^*$, typically an Empirical Risk Minimizer (ERM) on $S_1$. They then employed the decomposition

$$\mathbb{E}_{\rho}[L(h)] = \mathbb{E}_{\rho}[L(h) - L(h^*)] + L(h^*)$$

and used $S_2$ to give a PAC-Bayesian bound on $\mathbb{E}_{\rho}[L(h) - L(h^*)]$ and a single-hypothesis bound on $L(h^*)$. The quantity $\mathbb{E}_{\rho}[L(h) - L(h^*)]$ is known as *excess loss*. The advantage of this approach is that when $L(h^*)$ is a good approximation of $\mathbb{E}_{\rho}[L(h)]$, the excess loss has lower variance than the plain loss $\mathbb{E}_{\rho}[L(h)]$ and, therefore, is more efficient to bound, whereas the single-hypothesis bound on $L(h^*)$ does not involve the $\mathrm{KL}(\rho \| \pi)$ term. Therefore, it is generally beneficial to use excess losses in combination with data-informed priors. However, as with the previous approach, sacrificing $S_1$ to learn $\pi$ and $h^*$ means that the denominator in the bounds ($n$ in Theorem 1) reduces to the size of $S_2$, and it does not always pay off. (We note that the excess loss is not binary and not in the $[0, 1]$ interval, and in order to exploit small variance it is actually necessary to apply a PAC-Bayes-Empirical-Bernstein-style inequality (Tolstikhin and Seldin, 2013, Mhammedi et al., 2019, Wu et al., 2021) or the PAC-Bayes-split-kl inequality (Wu and Seldin, 2022) rather than Theorem 1, but the point about reduced sample size still applies.)

**Recursive PAC-Bayes (new)**  We introduce the following decomposition of the loss

$$\mathbb{E}_{\rho}[L(h)] = \mathbb{E}_{\rho}[L(h) - \gamma \mathbb{E}_{\pi}[L(h')]] + \gamma \mathbb{E}_{\pi}[L(h')]. \tag{1}$$

As before, we decompose $S$ into two disjoint sets $S = S_1 \cup S_2$. We make the following major observations:

- The quantity $\mathbb{E}_\pi[L(h')]$ on the right is "of the same kind" as $\mathbb{E}_\rho[L(h)]$ on the left.

- We can take an uninformed prior $\pi_0$ and apply Theorem 1 (or any other suitable PAC-Bayes bound) to bound $\mathbb{E}_\pi[L(h')]$. (The KL term in the bound on $\mathbb{E}_\pi[L(h')]$ will be $\mathrm{KL}(\pi\|\pi_0)$.)

- We can restrict $\pi$ to depend only on $S_1$, but still use all the data $S$ in calculation of the PAC-Bayes bound on $\mathbb{E}_\pi[L(h')]$, because $\pi$ is a posterior relative to $\pi_0$, and a posterior is allowed to depend on all the data, and in particular on any subset of the data. Therefore, the empirical loss $\mathbb{E}_\pi[\hat{L}(h', S)]$ can be computed on all the data $S$, and the denominator of the bound in Theorem 1 can be the size of $S$, and not the size of $S_2$. This is what we call *preservation of confidence information on $\pi$*, because all the data $S$ are used to construct a confidence bound on $\mathbb{E}_\pi[L(h')]$, and not just $S_2$. This is in contrast to the bound on $L(h^*)$ in the approach of Mhammedi et al. (2019), which only allows to use $S_2$ for bounding $L(h^*)$. Note that while we use all the data $S$ in calculation of the bound, we only use $S_1$ and $\mathbb{E}_\pi[\hat{L}(h', S_1)]$ in the construction of $\pi$. Nevertheless, we can still use the knowledge that we will have $n$ samples when we reach the estimation phase, i.e., when constructing $\pi$ we can leave the denominator of the bound at $n$, allowing more aggressive deviation from $\pi_0$.

- If we restrict $\pi$ to depend only on $S_1$, then it is a valid prior for estimation of any posterior quantity $\mathbb{E}_\rho[\cdot]$ based on $S_2$. Thus, if we also restrict $\gamma$ to depend only on $S_1$, we can use any PAC-Bayes-Empirical-Bernstein-style inequality or the PAC-Bayes-split-kl inequality to estimate the excess loss $\mathbb{E}_\rho[L(h) - \gamma\mathbb{E}_\pi[L(h')]]$ based on $S_2$, i.e., based on $\mathbb{E}_\rho[\hat{L}(h, S_2) - \gamma\mathbb{E}_\pi[\hat{L}(h', S_2)]]$. If $\gamma\mathbb{E}_\pi[L(h')]$ is a good approximation of $\mathbb{E}_\rho[L(h)]$ and $\mathbb{E}_\rho[L(h)]$ is not close to zero, then the excess loss $\mathbb{E}_\rho[L(h) - \gamma\mathbb{E}_\pi[L(h')]]$ is more efficient to bound than the plain loss $\mathbb{E}_\rho[L(h)]$.

- In general, since $\mathbb{E}_\rho[L(h)]$ is expected to improve on $\mathbb{E}_\pi[L(h')]$, it is natural to set $\gamma < 1$. However, $\gamma$ is not allowed to depend on $S_2$, because otherwise $\hat{L}(h, S_2) - \gamma\mathbb{E}_\pi[\hat{L}(h', S_2)]$ becomes a biased estimate of $L(h) - \gamma\mathbb{E}_\pi[L(h')]$. We discuss the choice of $\gamma$ in more detail when we present the bound and the experiments.

- Biggs and Guedj (2023) have proposed a sequential martingale-style evaluation of a martingale version of $\mathbb{E}_\rho[L(h) - L(h^*)]$ and $L(h^*)$ in the approach of Mhammedi et al., but it has not been shown to yield significant improvements yet. The same "martingalization" can be directly applied to our decomposition, but to keep things simple we stay with the basic decomposition.

- Finally, we note that we can split $S_1$ further and apply (1) recursively to bound $\mathbb{E}_\pi[L(h')]$.

To set notation for recursive decomposition, we use $\pi_0, \pi_1, \ldots, \pi_T$ to denote a sequence of distributions on $\mathcal{H}$, where $\pi_0$ is an uninformed prior and $\pi_T = \rho$ is the final posterior. We use $\gamma_2, \ldots, \gamma_T$ to denote a sequence of coefficients. For $t \geq 2$ we then have the recursive decomposition

$$\mathbb{E}_{\pi_t}[L(h)] = \mathbb{E}_{\pi_t}[L(h) - \gamma_t\mathbb{E}_{\pi_{t-1}}[L(h)]] + \gamma_t\mathbb{E}_{\pi_{t-1}}[L(h)]. \qquad (2)$$

To construct $\pi_1, \ldots, \pi_T$ we split the data $S$ into $T$ non-overlapping subsets, $S = S_1 \cup \cdots \cup S_T$. We restrict $\pi_t$ to depend on $U_t^{\mathrm{train}} = \bigcup_{s=1}^t S_s$ only, and we use $U_t^{\mathrm{val}} = \bigcup_{s=t}^T S_s$ to estimate (recursively) $\mathbb{E}_{\pi_t}[L(h)]$ (see Figures 1 and 2 in Appendix A for a graphical illustration). Note that $S_t$ is used both for construction of $\pi_t$ and for estimation of $\mathbb{E}_{\pi_t}[L(h)]$ (it is both in $U_t^{\mathrm{train}}$ and $U_t^{\mathrm{val}}$), resulting in efficient use of the data. It is possible to use any standard PAC-Bayes bound, e.g., Theorem 1, to bound $\mathbb{E}_{\pi_1}[L(h)]$, and any PAC-Bayes-Empirical-Bernstein-style bound or the PAC-Bayes-split-kl bound to bound the excess losses $\mathbb{E}_{\pi_t}[L(h) - \gamma_t\mathbb{E}_{\pi_{t-1}}[L(h)]]$. The excess losses take more than three values, so in the next section we present a generalization of the PAC-Bayes-split-kl inequality to general discrete random variables, which may be of independent interest. The Recursive PAC-Bayes bound is presented in Section 4.

## 3   Split-kl and PAC-Bayes-split-kl inequalities for discrete random variables

The kl inequality is one of the tightest concentration of measure inequalities for binary random variables. Letting $\mathrm{kl}^{-1,+}(\hat{p}, \varepsilon) := \max\{p : p \in [0, 1] \text{ and } \mathrm{kl}(\hat{p}\|p) \leq \varepsilon\}$ denote the upper inverse of kl and $\mathrm{kl}^{-1,-}(\hat{p}, \varepsilon) := \min\{p : p \in [0, 1] \text{ and } \mathrm{kl}(\hat{p}\|p) \leq \varepsilon\}$ the lower inverse, it states the following.

**Theorem 2** (kl Inequality (Langford, 2005, Foong et al., 2021, 2022)). *Let $Z_1, \cdots, Z_n$ be independent random variables bounded in the $[0,1]$ interval and with $\mathbb{E}[Z_i] = p$ for all $i$. Let $\hat{p} = \frac{1}{n}\sum_{i=1}^n Z_i$ be the empirical mean. Then, for any $\delta \in (0,1)$:*

$$\mathbb{P}\left(p \geq \mathrm{kl}^{-1,+}\left(\hat{p}, \frac{1}{n}\ln\frac{1}{\delta}\right)\right) \leq \delta \quad ; \quad \mathbb{P}\left(p \leq \mathrm{kl}^{-1,-}\left(\hat{p}, \frac{1}{n}\ln\frac{1}{\delta}\right)\right) \leq \delta.$$

While the kl inequality is tight for binary random variables, it is loose for random variables taking more than two values due to its inability to exploit small variance. To address this shortcoming Wu and Seldin (2022) have presented the split-kl and PAC-Bayes-split-kl inequalities for ternary random variables. Ternary random variables naturally appear in a variety of applications, including analysis of excess losses, certain ways of analysing majority votes, and in learning with abstention. The bound of Wu and Seldin is based on decomposition of a ternary random variable into a pair of binary random variables and application of the kl inequality to each of them. Their decomposition yields a tight bound in the binary and ternary case, but loose otherwise. The same decomposition was used by Biggs and Guedj (2023) to derive a slight variation of the inequality, with the same limitations. We present a novel decomposition of discrete random variables into a superposition of binary random variables. Unlike the decomposition of Wu and Seldin, which only applies in the ternary case, our decomposition applies to general discrete random variables. By combining it with kl bounds for the binary elements we obtain a tight bound. The decomposition is presented formally below and illustrated graphically in Figure 3 in Appendix A.

## 3.1 Split-kl inequality

Let $Z \in \{b_0, \ldots, b_K\}$ be a $(K+1)$-valued random variable with $b_0 < b_1 < \cdots < b_K$. For $j \in \{1, \ldots, K\}$ define $Z_{|j} = \mathbb{1}(Z \geq b_j)$ and $\alpha_j = b_j - b_{j-1}$. Then $Z = b_0 + \sum_{j=1}^K \alpha_j Z_{|j}$. For a sequence $Z_1, \ldots, Z_n$ of $(K+1)$-valued random variables with the same support, let $Z_{i|j} = \mathbb{1}(Z_i \geq b_j)$ denote the elements of binary decomposition of $Z_i$.

**Theorem 3** (Split-kl inequality for discrete random variables). *Let $Z_1, \ldots, Z_n$ be i.i.d. random variables taking values in $\{b_0, \ldots, b_K\}$ with $\mathbb{E}[Z_i] = p$ for all $i$. Let $\hat{p}_{|j} = \frac{1}{n}\sum_{i=1}^n Z_{i|j}$. Then for any $\delta \in (0,1)$:*

$$\mathbb{P}\left(p \geq b_0 + \sum_{j=1}^K \alpha_j\,\mathrm{kl}^{-1,+}\left(\hat{p}_{|j}, \frac{1}{n}\ln\frac{K}{\delta}\right)\right) \leq \delta.$$

*Proof.* Let $p_{|j} = \mathbb{E}[\hat{p}_{|j}]$, then $p = b_0 + \sum_{j=1}^K \alpha_j p_{|j}$ and

$$\mathbb{P}\left(p \geq b_0 + \sum_{j=1}^K \alpha_j\,\mathrm{kl}^{-1,+}\left(\hat{p}_{|j}, \frac{1}{n}\ln\frac{K}{\delta}\right)\right) \leq \mathbb{P}\left(\exists j : p_{|j} \geq \mathrm{kl}^{-1,+}\left(\hat{p}_{|j}, \frac{1}{n}\ln\frac{K}{\delta}\right)\right) \leq \delta,$$

where the first inequality is by the decomposition of $p$ and the second inequality is by the union bound and Theorem 2. □

## 3.2 PAC-Bayes-Split-kl inequality

Let $f : \mathcal{H} \times \mathcal{Z} \to \{b_0, \ldots, b_K\}$ be a $(K+1)$-valued loss function. (To connect it to the earlier examples, in the binary prediction case we would have $\mathcal{Z} = \mathcal{X} \times \mathcal{Y}$ with elements $Z = (X, Y)$ and $f(h, Z) = \ell(h(X), Y)$, but we will need a more general space $\mathcal{Z}$ later.) For $j \in \{1, \ldots, K\}$ let $f_{|j}(\cdot, \cdot) = \mathbb{1}(f(\cdot, \cdot) \geq b_j)$. Let $\mathcal{D}_Z$ be an unknown distribution on $\mathcal{Z}$. For $h \in \mathcal{H}$ let $F(h) = \mathbb{E}_{\mathcal{D}_Z}[f(h, Z)]$ and $F_{|j}(h) = \mathbb{E}_{\mathcal{D}_Z}[f_{|j}(h, Z)]$. Let $S = \{Z_1, \ldots, Z_n\}$ be an i.i.d. sample according to $\mathcal{D}_Z$ and $\hat{F}_{|j}(h, S) = \frac{1}{n}\sum_{i=1}^n f_{|j}(h, Z_i)$.

**Theorem 4** (PAC-Bayes-Split-kl Inequality). *For any distribution $\pi$ on $\mathcal{H}$ that is independent of $S$ and any $\delta \in (0,1)$:*

$$\mathbb{P}\left(\exists \rho \in \mathcal{P} : \mathbb{E}_\rho[F(h)] \geq b_0 + \sum_{j=1}^K \alpha_j\,\mathrm{kl}^{-1,+}\left(\mathbb{E}_\rho[\hat{F}_{|j}(h, S)], \frac{\mathrm{KL}(\rho\|\pi) + \ln\frac{2K\sqrt{n}}{\delta}}{n}\right)\right) \leq \delta,$$

*where $\mathcal{P}$ is the set of all possible probability distributions on $\mathcal{H}$ that can depend on $S$.*

*Proof.* We have $f(\cdot,\cdot) = b_0 + \sum_{j=1}^{K} \alpha_j f_{|j}(\cdot,\cdot)$ and $F(h) = b_0 + \sum_{j=1}^{K} \alpha_j F_{|j}(h)$. Therefore,

$$\mathbb{P}\left(\exists \rho \in \mathcal{P} : \mathbb{E}_\rho[F(h)] \geq b_0 + \sum_{j=1}^{K} \alpha_j \, \mathrm{kl}^{-1,+}\left(\mathbb{E}_\rho[\hat{F}_{|j}(h,S)], \frac{\mathrm{KL}(\rho\|\pi) + \ln\frac{2K\sqrt{n}}{\delta}}{n}\right)\right)$$

$$\leq \mathbb{P}\left(\exists \rho \in \mathcal{P} \text{ and } \exists j : \mathbb{E}_\rho[F_{|j}(h)] \geq \mathrm{kl}^{-1,+}\left(\mathbb{E}_\rho[\hat{F}_{|j}(h,S)], \frac{\mathrm{KL}(\rho\|\pi) + \ln\frac{2K\sqrt{n}}{\delta}}{n}\right)\right) \leq \delta,$$

where the first inequality is by the decomposition of $F$ and the second inequality is by the union bound and application of Theorem 1 to $F_{|j}$ (note that $f_{|j}$ is a zero-one loss function). $\qquad\square$

## 4 Recursive PAC-Bayes bound

Now we derive a Recursive PAC-Bayes bound based on the loss decomposition in equation (2). We aim to bound $\mathbb{E}_{\pi_t}[L(h) - \gamma_t \mathbb{E}_{\pi_{t-1}}[L(h')]]$, which we denote by

$$F_{\gamma_t,\pi_{t-1}}(h) = L(h) - \gamma_t \mathbb{E}_{\pi_{t-1}}[L(h')] = \mathbb{E}_{\mathcal{D}\times\pi_{t-1}}[\ell(h(X),Y) - \gamma_t \ell(h'(X),Y)],$$

where $\mathcal{D} \times \pi_{t-1}$ is a product distribution on $\mathcal{X} \times \mathcal{Y} \times \mathcal{H}$ and $h' \in \mathcal{H}$ is sampled according to $\pi_{t-1}$. We further define

$$f_{\gamma_t}(h,(X,Y,h')) = \ell(h(X),Y) - \gamma_t \ell(h'(X),Y) \in \{-\gamma_t, 0, 1-\gamma_t, 1\},$$

then $F_{\gamma_t,\pi_{t-1}}(h) = \mathbb{E}_{\mathcal{D}\times\pi_{t-1}}[f_{\gamma_t}(h,(X,Y,h'))]$. In order to apply Theorem 4, we represent $f_{\gamma_t}$ as a superposition of binary functions. For this purpose we let $\{b_{t|0}, b_{t|1}, b_{t|2}, b_{t|3}\} = \{-\gamma_t, 0, 1-\gamma_t, 1\}$ and define $f_{\gamma_t|j}(h,(X,Y,h')) = \mathbb{1}(f_{\gamma_t}(h,(X,Y,h')) \geq b_{t|j})$. We let $F_{\gamma_t,\pi_{t-1}|j}(h) = \mathbb{E}_{\mathcal{D}\times\pi_{t-1}}[f_{\gamma_t|j}(h,(X,Y,h'))]$, then $F_{\gamma_t,\pi_{t-1}}(h) = -\gamma_t + \sum_{j=1}^{3}(b_{t|j} - b_{t|j-1})F_{\gamma_t,\pi_{t-1}|j}(h)$.

Now we construct an empirical estimate of $F_{\gamma_t,\pi_{t-1}|j}(h)$. We first let $\hat{\pi}_{t-1} = \{h_1^{\pi_{t-1}}, h_2^{\pi_{t-1}}, \dots\}$ be a sequence of prediction rules sampled independently according to $\pi_{t-1}$. We define $U_t^{\mathrm{val}} \circ \hat{\pi}_{t-1} = \{(X_i, Y_i, h_i^{\pi_{t-1}}) : (X_i, Y_i) \in U_t^{\mathrm{val}}\}$. In words, for every sample $(X_i, Y_i) \in U_t^{\mathrm{val}}$ we sample a prediction rule $h_i^{\pi_{t-1}}$ according to $\pi_{t-1}$ and put the triplet $(X_i, Y_i, h_i^{\pi_{t-1}})$ in $U_t^{\mathrm{val}} \circ \hat{\pi}_{t-1}$. The triplets $(X_i, Y_i, h_i^{\pi_{t-1}})$ correspond to the random variables $Z$ in Theorem 4. We note that $|U_t^{\mathrm{val}}| = |U_t^{\mathrm{val}} \circ \hat{\pi}_{t-1}|$, and we let $n_t^{\mathrm{val}} = |U_t^{\mathrm{val}}|$. We define the empirical estimate of $F_{\gamma_t,\pi_{t-1}|j}(h)$ as $\hat{F}_{\gamma_t|j}(h, U_t^{\mathrm{val}} \circ \hat{\pi}_{t-1}) = \frac{1}{n_t^{\mathrm{val}}} \sum_{(X,Y,h')\in U_t^{\mathrm{val}}\circ\hat{\pi}_{t-1}} f_{\gamma_t|j}(h,(X,Y,h'))$. Note that $\mathbb{E}_{\mathcal{D}\times\pi_{t-1}}[\hat{F}_{\gamma_t|j}(h, U_t^{\mathrm{val}} \circ \hat{\pi}_{t-1})] = F_{\gamma_t,\pi_{t-1}|j}(h)$, therefore, we can use Theorem 4 to bound $\mathbb{E}_{\pi_t}[F_{\gamma_t,\pi_{t-1}}(h)]$ using its empirical estimates. We are now ready to state the bound.

**Theorem 5** (Recursive PAC-Bayes Bound). *Let $S = S_1 \cup \dots \cup S_T$ be an i.i.d. sample split in an arbitrary way into $T$ non-overlapping subsamples, and let $U_t^{\mathrm{train}} = \bigcup_{s=1}^{t} S_s$ and $U_t^{\mathrm{val}} = \bigcup_{s=t}^{T} S_s$. Let $n_t^{\mathrm{val}} = |U_t^{\mathrm{val}}|$. Let $\pi_0^*, \pi_1^*, \dots, \pi_T^*$ be a sequence of distributions on $\mathcal{H}$, where $\pi_t^*$ is allowed to depend on $U_t^{\mathrm{train}}$, but not the rest of the data. Let $\gamma_2, \dots, \gamma_T$ be a sequence of coefficients, where $\gamma_t$ is allowed to depend on $U_{t-1}^{\mathrm{train}}$, but not the rest of the data. For $t \in \{1, \dots, T\}$ let $\mathcal{P}_t$ be a set of distributions on $\mathcal{H}$, which are allowed to depend on $U_t^{\mathrm{train}}$. Then for any $\delta \in (0,1)$:*

$$\mathbb{P}(\exists t \in \{1,\dots,T\} \text{ and } \pi_t \in \mathcal{P}_t : \mathbb{E}_{\pi_t}[L(h)] \geq \mathrm{B}_t(\pi_t)) \leq \delta,$$

*where $\mathrm{B}_t(\pi_t)$ is a PAC-Bayes bound on $\mathbb{E}_{\pi_t}[L(h)]$ defined recursively as follows. For $t = 1$*

$$\mathrm{B}_1(\pi_1) = \mathrm{kl}^{-1,+}\left(\mathbb{E}_{\pi_1}[\hat{L}(h,S)], \frac{\mathrm{KL}(\pi_1\|\pi_0^*) + \ln\frac{2T\sqrt{n}}{\delta}}{n}\right).$$

*For $t \geq 2$ we let $\mathcal{E}_t(\pi_t, \gamma_t)$ denote a PAC-Bayes bound on $\mathbb{E}_{\pi_t}[L(h) - \gamma_t \mathbb{E}_{\pi_{t-1}^*}[L(h')]]$ given by*

$$\mathcal{E}_t(\pi_t, \gamma_t) = -\gamma_t + \sum_{j=1}^{3}(b_{t|j} - b_{t|j-1})\,\mathrm{kl}^{-1,+}\left(\mathbb{E}_{\pi_t}\left[\hat{F}_{\gamma_t|j}(h, U_t^{\mathrm{val}} \circ \hat{\pi}_{t-1}^*)\right], \frac{\mathrm{KL}(\pi_t\|\pi_{t-1}^*) + \ln\frac{6T\sqrt{n_t^{\mathrm{val}}}}{\delta}}{n_t^{\mathrm{val}}}\right)$$

*and then*

$$\mathrm{B}_t(\pi_t) = \mathcal{E}_t(\pi_t, \gamma_t) + \gamma_t \mathrm{B}_{t-1}(\pi_{t-1}^*). \tag{3}$$

*Proof.* By Theorem 1 we have $\mathbb{P}(\exists \pi_1 \in \mathcal{P}_1 : \mathbb{E}_{\pi_1}[L(h)] \geq B_1(\pi_1)) \leq \frac{\delta}{T}$. Further, by Theorem 4 for $t \in \{2, \ldots, T\}$ we have $\mathbb{P}\left(\exists \pi_t \in \mathcal{P}_t : \mathbb{E}_{\pi_t}[L(h) - \gamma_t \mathbb{E}_{\pi_{t-1}^*}[L(h')]] \geq \mathcal{E}_t(\pi_t, \gamma_t)\right) \leq \frac{\delta}{T}$. The theorem follows by a union bound and the recursive decomposition of the loss (2). $\square$

**Discussion**

- Note that $\pi_1^*, \ldots, \pi_T^*$ can be constructed sequentially, but $\pi_t^*$ can only be constructed based on the data in $U_t^{\text{train}}$, meaning that in the construction of $\pi_t^*$ we can only rely on $\mathbb{E}_{\pi_t}\left[\hat{F}_{\gamma_t|j}(h, S_t \circ \hat{\pi}_{t-1})\right]$, but not on $\mathbb{E}_{\pi_t}\left[\hat{F}_{\gamma_t|j}(h, U_t^{\text{val}} \circ \hat{\pi}_{t-1})\right]$. Also note that $S_t$ is part of both $U_t^{\text{train}}$ and $U_t^{\text{val}}$ (see Figure 1 in Appendix A for a graphical illustration). In other words, when we evaluate the bounds we can use additional data. And even though the additional data can only be used in the evaluation stage, we can still use the knowledge that we will get more data for evaluation when we construct $\pi_t^*$. For example, we can take

$$\pi_1^* = \arg\min_\pi \text{kl}^{-1,+}\left(\mathbb{E}_\pi[\hat{L}(h, S_1)], \frac{\text{KL}(\pi\|\pi_0^*) + \ln\frac{2T\sqrt{n}}{\delta}}{n}\right) \tag{4}$$

and for $t \geq 2$

$$\pi_t^* = \arg\min_\pi \sum_{j=1}^{3}(b_{t|j} - b_{t|j-1})\,\text{kl}^{-1,+}\left(\mathbb{E}_\pi\left[\hat{F}_{\gamma_t|j}(h, S_t \circ \hat{\pi}_{t-1}^*)\right], \frac{\text{KL}(\pi\|\pi_{t-1}^*) + \ln\frac{6T\sqrt{n_t^{\text{val}}}}{\delta}}{n_t^{\text{val}}}\right). \tag{5}$$

The empirical losses above are calculated on $S_t$ corresponding to $\pi_t^*$, but the sample sizes $n_t^{\text{val}}$ correspond to the size of the validation set $U_t^{\text{val}}$ rather than the size of $S_t$. This allows to be more aggressive in deviating with $\pi_t^*$ from $\pi_{t-1}^*$ by sustaining larger $\text{KL}(\pi_t^*\|\pi_{t-1}^*)$ terms.

- Similarly, $\gamma_2, \ldots, \gamma_T$ can also be constructed sequentially, as long as $\gamma_t$ only depends on $U_{t-1}^{\text{train}}$ (otherwise $\hat{F}_{\gamma_t|j}(h, S_t \circ \hat{\pi}_{t-1}^*)$ becomes a biased estimate of $F_{\gamma_t, \pi_{t-1}^*|j}(h)$).

- We naturally want to have improvement over recursion steps, meaning $\text{B}_t(\pi_t^*) < \text{B}_{t-1}(\pi_{t-1}^*)$. Plugging this into (3), we obtain $\mathcal{E}(\pi_t^*, \gamma_t) + \gamma_t B_{t-1}(\pi_{t-1}^*) < B_{t-1}(\pi_{t-1}^*)$, which implies that we want $\gamma_t$ to be sufficiently small to satisfy $\gamma_t < 1 - \frac{\mathcal{E}_t(\pi_t^*, \gamma_t)}{B_{t-1}(\pi_{t-1}^*)}$. At the same time, $\gamma_t$ should be non-negative. Therefore, improvement over recursion steps can only be maintained as long as $\mathcal{E}_t(\pi_t^*, \gamma_t) < B_{t-1}(\pi_{t-1}^*)$. We note that $\gamma_t\,\text{B}_{t-1}(\pi_{t-1}^*)$ term in (3) is linearly increasing in $\gamma_t$, whereas $\mathcal{E}(\pi_t^*, \gamma_t)$ is decreasing in $\gamma_t$. The value of $\gamma_t$ that minimizes the trade-off depends on the data. Even though it is not allowed to use $U_t^{\text{val}}$ for tuning $\gamma_t$, it is possible to take a grid of values of $\gamma_t$ and a union bound over the grid, and then select the best value from the grid based the value of the bound evaluated on $U_t^{\text{val}}$.

## 5 Experiments

In this section, we provide an empirical comparison of our Recursive PAC-Bayes (RPB) procedure to the following prior work: i) Uninformed priors (Uninformed), (Dziugaite and Roy, 2017); ii) Data-informed priors (Informed) (Ambroladze et al., 2007, Pérez-Ortiz et al., 2021); iii) Data-informed prior + excess loss (Informed + Excess) (Mhammedi et al., 2019, Wu and Seldin, 2022). All the experiments were run on a laptop. The source code for replicating the experiments is available at Github[1].

We start with describing the details of the optimization procedure, and then present the results.

### 5.1 Details of the optimization and evaluation procedure

We constructed $\pi_1^*, \ldots, \pi_T^*$ sequentially using the optimization objective, (4) for $\pi_1^*$ and (5) for $\pi_2^*$ to $\pi_T^*$, and computed the bound using the recursive procedure in Theorem 5. There are a few technical details concerning convexity of the optimization procedure and infinite size of the set of prediction rules $\mathcal{H}$ that we address next.

---

[1]https://github.com/pyijiezhang/rpb

### 5.1.1 Convexification of the loss functions

The functions $f_{\gamma_t|j}(h, (X, Y, h'))$ defined in Section 4 are non-convex and non-differentiable: $f_{\gamma_t|j}(h, (X, Y, h')) = \mathbb{1}\left(f_{\gamma_t}(h, (X, Y, h')) \geq b_{t|j}\right) = \mathbb{1}\left(\ell(h(X), Y) - \gamma_t \ell(h'(X), Y) \geq b_{t|j}\right)$. In order to facilitate optimization, we approximate the external indicator function $\mathbb{1}(z \geq z_0)$ by a sigmoid function $\omega(z; c_1, z_0) = (1 + \exp(c_1(z - z_0)))^{-1}$ with a fixed parameter $c_1 > 0$ specified in Appendix B.3.

Furthermore, since the zero-one loss $\ell(h(X), Y)$ is also non-differentiable, we adopt the cross-entropy loss, as in most modern training procedures (Pérez-Ortiz et al., 2021). Specifically, for a $k$-class classification problem, let $h : \mathcal{X} \to \mathbb{R}^k$ represent the function implemented by the neural network, assigning each class a real value. Let $u = h(X)$ be the assignment, with $u_i$ being the $i$-th value of the vector. To convert this real-valued vector into a probability distribution over classes, we apply the softmax function $\sigma : \mathbb{R}^k \to \Delta^{k-1}$, where $\sigma(u)_i = \exp(c_2 u_i) / \sum_j \exp(c_2 u_j)$ for some $c_2 > 0$ for each entry. The cross-entropy loss $\ell^{\text{ce}} : \mathbb{R}^k \times [k] \to \mathbb{R}$ is defined by $\ell^{\text{ce}}(u, Y) = -\log(\sigma(u)_Y)$. However, since this loss is unbounded, whereas the PAC-Bayes-kl bound requires losses within $[0, 1]$, we enforce a $[0, 1]$-valued cross-entropy loss by mixing the output distribution with a uniform distribution $\sigma(u)$, i.e., $\tilde{\sigma}(u)_i = (1 - p_{\min})\sigma(u)_i + p_{\min}/k$ for all $i \in [k]$ and for some $p_{\min} > 0$, and then rescaling it to $[0, 1]$ by taking $\tilde{\ell}^{\text{ce}}(u, Y) = -\log(\tilde{\sigma}(u)_Y)/\log(k/p_{\min})$.

We emphasize that in the evaluation of the bound (using Theorem 5), we directly compute the zero-one loss and the $f_{\gamma_t|j}$ functions without employing the approximations.

### 5.1.2 Relaxation of the PAC-Bayes-kl bound

The PAC-Bayes-kl bound is often criticized for being unfriendly to optimization (Rodríguez-Gálvez et al., 2024). Therefore, several relaxations have been proposed, including the PAC-Bayes-classic bound (McAllester, 1999), the PAC-Bayes-$\lambda$ bound (Thiemann et al., 2017), and the PAC-Bayes-quadratic bound (Rivasplata et al., 2019, Pérez-Ortiz et al., 2021), among others. In our optimization we have adopted the bound of McAllester (1999) instead of the kl-based bounds in Equation (5).

We again emphasize that in the evaluation of the bound we used the kl-based bounds in Theorem 5.

### 5.1.3 Estimation of $\mathbb{E}_\pi[\cdot]$

Due to the infinite size of $\mathcal{H}$ and lack of a closed-form expression for $\mathbb{E}_{\pi_1}[\hat{L}(h, S)]$ and $\mathbb{E}_{\pi_t}[\hat{F}_{\gamma_t|j}(h, U_t^{\text{val}} \circ \hat{\pi}_{t-1}^*)]$ appearing in Theorem 5, we approximate them by sampling (Pérez-Ortiz et al., 2021). For optimization, we sample one classifier for each mini-batch during stochastic gradient descent. For evaluation, we sample one classifier for each data in the corresponding evaluation dataset. Due to approximation of the empirical quantities the final bound in Theorem 5 requires an additional concentration bound. (We note that the extra bound is only required for computation of the final bound, but not for optimization of $\hat{\pi}_t^*$.) Specifically, let $\hat{\pi}_t^* = \{h_1^{\pi_t}, h_2^{\pi_t}, \ldots, h_m^{\pi_t}\}$ be $m$ prediction rules sampled independently according to $\pi_t$. Then for any function $f(h)$ taking values in $[0, 1]$ (which is the case for $\hat{L}(h, S)$ and $\hat{F}_{\gamma_t|j}(h, U_t^{\text{val}} \circ \hat{\pi}_{t-1}^*)$) and $\delta' \in (0, 1)$ we have

$$\mathbb{P}\left(\mathbb{E}_{\pi_t^*}[f(h)] \geq \text{kl}^{-1,+}\left(\frac{1}{m} \sum_{i=1}^m f(h_i^{\pi_t^*}), \frac{1}{m} \log \frac{1}{\delta'}\right)\right) \leq \delta'.$$

It is worth noting that $\mathbb{E}_{\pi_t^*}[f(h)]$ is evaluated for a fixed $\pi_t^*$, meaning that there is no selection involved, and therefore no KL term appears in the bound above. We, of course, take a union bound over all the quantities being estimated.

## 5.2 Experimental results

We evaluated our approach and compared it to prior work using multi-class classification tasks on MNIST (LeCun and Cortes, 2010) and Fashion MNIST (Xiao et al., 2017) datasets, both with 60000 training data. The experimental setup was based on the work of Dziugaite and Roy (2017) and Pérez-Ortiz et al. (2021). Similar to them we used Gaussian distributions for all the priors and posteriors, modeled by probabilistic neural networks. Technical details are provided in B.

The empirical evaluation is presented in Table 1. For the Uninformed approach, we trained and evaluated the bound using the entire training dataset directly. For the other two baseline methods, Informed and Informed + Excess Loss, we used half of the training data to train the informed prior and an ERM $h^*$ for the excess loss, and the other half to learn the posterior. For our Recursive PAC-Bayes, we chose $\gamma_t = 1/2$ for all $t$, and conducted experiments with $T = 2, 4, 6, 8$ to study the impact of recursion depth. (Each value of $T$ corresponded to a separate run of the algorithm and a separate evaluation of the bound, i.e., they should not be seen as successive refinements.) We applied a geometric data split. Specifically, for $T = 2$ the split was (30000, 30000) points; for $T = 4$, it was (7500, 7500, 15000, 30000); for $T = 6$ it was (1875, 1875, 3750, 7500, 15000, 30000); and for $T = 8$, it was (469, 469, 937, 1875, 3750, 7500, 15000, 30000). This approach allowed the early recursion steps, which had fewer data points, to efficiently learn the prior, while preserving enough data for fine-tuning in the later steps. Note that with this approach the value of $n_t^{\text{val}} = |U_t^{\text{val}}| = \sum_{s=t}^{T} |S_s|$, which is in the denominator of the bounds in Theorem 5, is at least $\frac{n}{2}$.

Table 1: Comparison of the classification loss of the final posterior $\rho$ on the entire training data, $\mathbb{E}_\rho[\hat{L}(h, S)]$ (Train 0-1), and on the testing data, $\mathbb{E}_\rho[\hat{L}(h, S^{\text{test}})]$ (Test 0-1), and the corresponding bounds for each method on MNIST and Fashion MNIST. We report the mean and one standard deviation over 5 repetitions. "Unif." abbreviates the Uniform approach, "Inf." the Informed, "Inf. + Ex." the Informed + Excess Loss, and "RPB" the Recursive PAC-Bayes.

| | MNIST | | | Fashion MNIST | | |
|---|---|---|---|---|---|---|
| | Train 0-1 | Test 0-1 | Bound | Train 0-1 | Test 0-1 | Bound |
| Uninf. | .343 (2e-3) | .335 (3e-3) | .457 (2e-3) | .382 (2e-3) | .384 (2e-3) | .464 (2e-3) |
| Inf. | .377 (8e-4) | .371 (6e-3) | .408 (9e-4) | .412 (1e-3) | .413 (6e-3) | .440 (1e-3) |
| Inf. + Ex. | .157 (2e-3) | .151 (3e-3) | .192 (2e-3) | .280 (4e-3) | .285 (5e-3) | .342 (6e-3) |
| RPB $T = 2$ | .143 (2e-3) | .139 (3e-3) | .321 (3e-3) | .257 (3e-3) | .266 (5e-3) | .404 (3e-3) |
| RPB $T = 4$ | .112 (1e-3) | .109 (1e-3) | .203 (8e-4) | .203 (2e-3) | .213 (3e-3) | .293 (1e-3) |
| RPB $T = 6$ | .103 (1e-3) | .101 (1e-3) | .166 (1e-3) | .186 (4e-4) | .198 (1e-3) | .255 (1e-3) |
| RPB $T = 8$ | **.101 (1e-3)** | **.097 (2e-3)** | **.158 (2e-3)** | **.181 (1e-3)** | **.192 (3e-3)** | **.242 (1e-3)** |

Table 1 shows that even with only $T = 2$, which corresponds to the data split used in the Informed and the Informed + Excess Loss approaches, RPB achieves better test performance than prior work. As the recursion deepens, further improvements in both the test error and the bound are observed. We note that while the bound for $T = 2$ is looser compared to the Informed + Excess Loss method, deeper recursion yields bounds that are tighter. Overall, deep recursion provides substantial improvements in the bound and the test error relative to prior work.

Tables 2 and 3 provide a glimpse into the training progress of RPB with $T = 8$ by showing the evolution of the key quantities along the recursive process. Similar tables for other values of $T$ are provided in Appendix B.4, along with training details for other methods. The tables show an impressive reduction of the KL term and significant improvement of the bound as the recursion proceeds, demonstrating effectiveness of the approach.

Table 2: Insight into the training process of the Recursive PAC-Bayes for $T = 8$ on MNIST. The table shows the evolution of $\mathcal{E}_t(\pi_t^*, \gamma_t)$, $B_t(\pi_t^*)$, and other quantities as the training progresses with $t$. We define $\hat{F}_{\gamma_t}(h, U_t^{\text{val}} \circ \hat{\pi}_{t-1}) = -\gamma_t + \sum_{j=1}^{3}(b_{t|j} - b_{t|j-1})\hat{F}_{\gamma_t|j}(h, U_t^{\text{val}} \circ \hat{\pi}_{t-1})$.

| $t$ | $n_t^{\text{val}}$ | $\mathbb{E}_{\pi_t}[\hat{F}_{\gamma_t}(h, U_t^{\text{val}} \circ \hat{\pi}_{t-1})]$ | $\frac{\text{KL}(\pi_t^* \| \pi_{t-1}^*)}{n_t^{\text{val}}}$ | $\mathcal{E}_t(\pi_t^*, \gamma_t)$ | $B_t(\pi_t^*)$ | Test 0-1 |
|---|---|---|---|---|---|---|
| 1 | 60000 | | .009 (3e-4) | | .612 (9e-3) | .532 (.011) |
| 2 | 59532 | -0.046 (4e-3) | .031 (1e-3) | .114 (2e-3) | .421 (5e-3) | .215 (7e-3) |
| 3 | 59063 | .040 (3e-3) | .013 (9e-4) | .125 (3e-3) | .336 (2e-3) | .146 (3e-3) |
| 4 | 58125 | .049 (1e-3) | .005 (3e-4) | .099 (1e-3) | .267 (7e-4) | .120 (2e-3) |
| 5 | 56250 | .052 (4e-4) | .002 (1e-4) | .083 (1e-3) | .217 (1e-3) | .111 (2e-3) |
| 6 | 52500 | .051 (1e-3) | .001 (4e-5) | .076 (1e-3) | .185 (1e-3) | .104 (2e-3) |
| 7 | 45000 | .050 (1e-3) | 8e-4 (6e-5) | .073 (1e-3) | .166 (1e-3) | .099 (1e-3) |
| 8 | 30000 | .050 (1e-3) | 6e-4 (4e-5) | .074 (1e-3) | .158 (2e-3) | .097 (2e-3) |

Table 3: Insight into the training process of the Recursive PAC-Bayes for $T = 8$ on Fashion MNIST.

| $t$ | $n_t^{\text{val}}$ | $\mathbb{E}_{\pi_t}[\hat{F}_{\gamma_t}(h, U_t^{\text{val}} \circ \hat{\pi}_{t-1})]$ | $\frac{\text{KL}(\pi_t^*\|\pi_{t-1}^*)}{n_t^{\text{val}}}$ | $\mathcal{E}_t(\pi_t^*, \gamma_t)$ | $B_t(\pi_t^*)$ | Test 0-1 |
|---|---|---|---|---|---|---|
| 1 | 60000 | | .003 (7e-5) | | .733 (7e-3) | .686 (8e-3) |
| 2 | 59532 | -0.043 (8e-3) | .023 (6e-4) | .104 (9e-3) | .470 (8e-3) | .309 (7e-3) |
| 3 | 59063 | .083 (4e-3) | .008 (3e-4) | .161 (3e-3) | .396 (4e-3) | .242 (1e-3) |
| 4 | 58125 | .090 (3e-3) | .004 (5e-4) | .142 (4e-3) | .341 (5e-4) | .216 (5e-3) |
| 5 | 56250 | .093 (3e-3) | .001 (2e-4) | .126 (3e-3) | .297 (4e-3) | .204 (4e-3) |
| 6 | 52500 | .090 (1e-3) | 6e-4 (6e-5) | .117 (1e-3) | .265 (1e-3) | .195 (3e-3) |
| 7 | 45000 | .090 (1e-3) | 4e-4 (2e-5) | .115 (1e-3) | .248 (1e-3) | .195 (5e-4) |
| 8 | 30000 | .090 (1e-3) | 4e-4 (1e-5) | .117 (1e-3) | .242 (1e-3) | .192 (3e-3) |

## 6  Discussion

We have presented the first PAC-Bayesian bound that supports sequential prior updates and preserves confidence information on the prior. The work closes a long-standing gap between Bayesian and Frequentist learning by making sequential data processing and sequential updates of prior knowledge meaningful and beneficial in the frequentist framework, as it has always been in the Bayesian framework. We have shown that apart from theoretical beauty the approach is highly beneficial in practice.

The Recursive PAC-Bayes framework is extremely rich and powerful, and leads to numerous directions for future research, some of which we briefly sketch next.

- The decomposition in (2) applies to any loss function, including unbounded losses. It would be interesting to find additional applications to it.

- While we have restricted ourselves to the zero-one loss function to illustrate the use of PAC-Bayes-split-kl, the results can be directly generalized to any bounded loss function by replacing PAC-Bayes-split-kl with PAC-Bayes-Empirical-Bernstein or PAC-Bayes-Unexpected-Bernstein, and deriving the corresponding analogue of Theorem 5 (which is straightforward).

- We have shown that the bound works well with geometric split of the data, but there are many other ways to split the data which could be studied.

- There is also a lot of space for experimentation with optimization of $\gamma_t$.

- It would be interesting to study how the bound will perform in sequential learning settings, where the data arrives sequentially, and thus the partition is dictated externally.

- There are many interesting research directions from the computational perspective. We note that for base models with linear computational complexity (e.g., neural networks) the overhead of recursion is relatively small and optimization time of Recursive PAC-Bayes is comparable to processing all data at once or in two chunks (as in data-dependent priors). For base models with superlinear computational complexity (e.g., kernel SVMs) sequential training of several small models in the recursion may actually be cheaper than training a big model based on all the data. Moreover, since the bound in Theorem 5 holds for any sequence of distributions $\pi_0^*, \pi_1^*, \ldots, \pi_T^*$, the optimization in equation (5) is allowed to be approximate. Considering that the improvement of the bounds and the test loss relative to prior work was very significant, there is space to look at the trade-off between statistical power and computational complexity. Namely, it may potentially be possible to relax the approximation of $\arg\min$ in equation (5) to gain computational speed-up at the cost of only a small compromise on the bounds and test losses.

- We note that it is possible to start the recursion at $\pi_0$. Namely, it is possible to use, for example, Theorem 2 to bound $\mathbb{E}_{\pi_0}[L(h)]$ using all the data, and apply the recursive decomposition (2) starting from $\pi_1$. Whether this would yield an advantage relative to starting the recursion at $\pi_1$, as we did, remains to be studied.

## Acknowledgments and Disclosure of Funding

YW acknowledges support from the Novo Nordisk Foundation, grant number NNF21OC0070621. YZ acknowledge Ph.D. funding from Novo Nordisk A/S. BECA acknowledges funding from the ANR grant project BACKUP ANR-23-CE40-0018-01.

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

# A  Illustrations

In this appendix we provide graphical illustrations of the basic concepts presented in the paper.

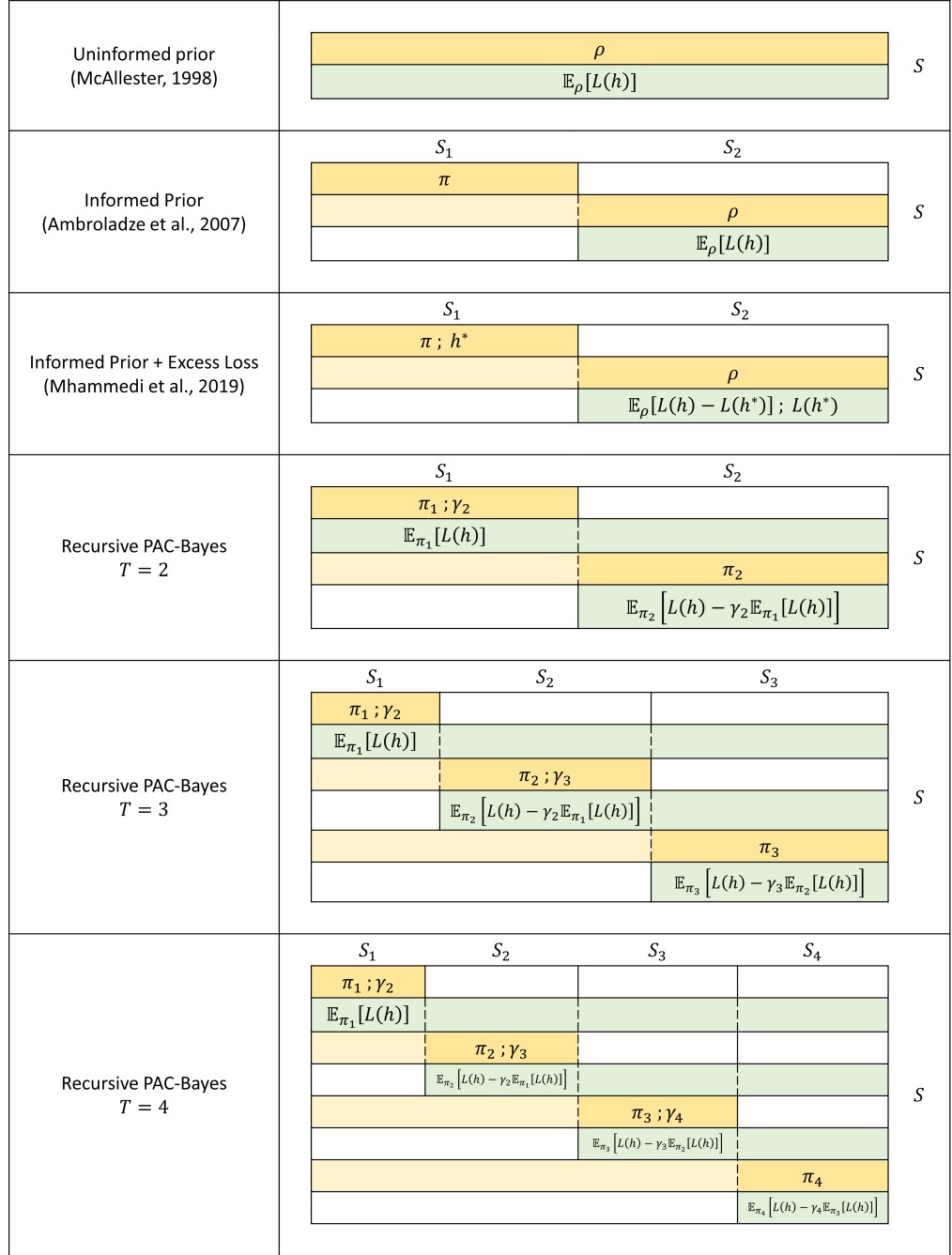

Figure 1: **Evolution of PAC-Bayes.** The figure shows how data are used by different PAC-Bayes approaches. Dark yellow shows data used directly for optimization of the indicated quantities. Light yellow shows data involved indirectly through dependence on the prior. Light green shows data used for estimation of the indicated quantities. In Recursive PAC-Bayes data are released and used sequentially chunk-by-chunk, as indicated by the dashed lines. For example, in the $T = 4$ case $\mathbb{E}_{\pi_1}[L(h)]$ is first evaluated on $S_1$ to construct $\pi_1$ and $\gamma_2$, then in the first recursion step on $S_1 \cup S_2$, in the second step on $S_1 \cup S_2 \cup S_3$, and in the last step on all $S$.

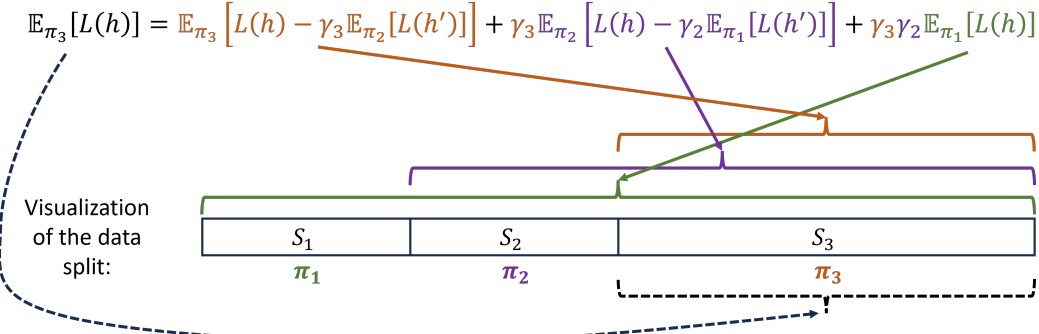

Figure 2: **Recursive Decomposition into Three Terms.** The figure illustrates recursive decomposition of $\mathbb{E}_{\pi_3}[L(h)]$ into three terms based on equation (2), and a geometric data split, as used in our experiments. The bottom line illustrates which data are used for construction of which distribution: $S_1$ for $\pi_1$; $S_2$ for $\pi_2$; and $S_3$ for $\pi_3$. The brackets above the data show which data are used for computing PAC-Bayes bounds for which term: $S_1 \cup S_2 \cup S_3$ for $\mathbb{E}_{\pi_1}[L(h)]$; $S_2 \cup S_3$ for $\mathbb{E}_{\pi_2}[L(h) - \gamma_2 \mathbb{E}_{\pi_1}[L(h')]]$; and $S_3$ for $\mathbb{E}_{\pi_3}[L(h) - \gamma_3 \mathbb{E}_{\pi_2}[L(h')]]$. Note that a direct computation of a PAC-Bayes bound on $\mathbb{E}_{\pi_3}[L(h)]$ would have only allowed to use the data in $S_3$, as shown by the black dashed line. The figure illustrates that recursive decomposition provides more efficient use of the data. We also note that initially we start with poor priors, and so the $\mathrm{KL}(\pi_t \| \pi_{t-1})$ term for small $t$ is expected to be large, but this is compensated by a small multiplicative factor $\prod_{i=t+1}^{T} \gamma_i$ and availability of a lot of data $\bigcup_{i=t}^{T} S_i$ for computing the PAC-Bayes bound. For example, $\mathbb{E}_{\pi_1}[L(h)]$ is multiplied by $\gamma_3 \gamma_2$ and we can use all the data for computing a PAC-Bayes bound on this term. By the time we reach higher $t$, the priors $\pi_{t-1}$ get better, and the $\mathrm{KL}(\pi_t \| \pi_{t-1})$ term in the bounds gets much smaller, and additionally the bounds benefit from the small variance of the excess loss. With geometric split of the data, we use little data to quickly move $\pi_t$ to a good region, and then we still have enough data for a good estimation of the later terms, like $\mathbb{E}_{\pi_3}[L(h) - \gamma_3 \mathbb{E}_{\pi_2}[L(h')]]$.

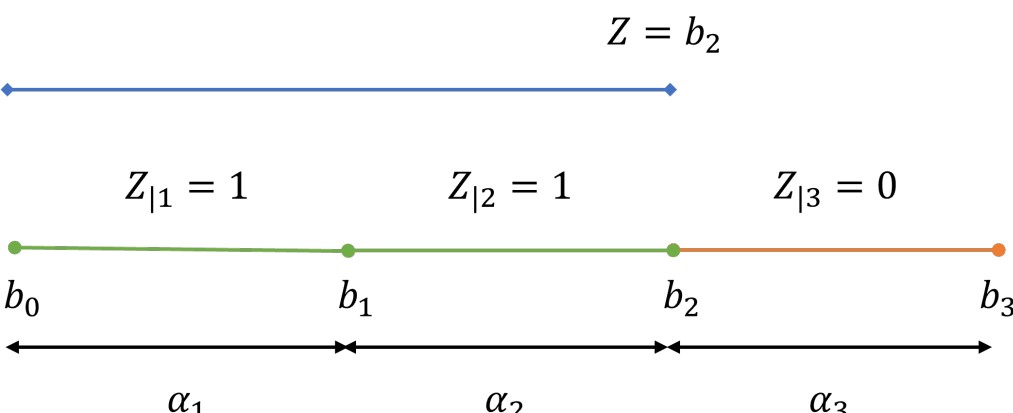

Figure 3: **Decomposition of a discrete random variable into a superposition of binary random variables.** The figure illustrates a decomposition of a discrete random variable $Z$ with domain of four values $b_0 < b_1 < b_2 < b_3$ into a superposition of three binary random variables, $Z = b_0 + \sum_{j=1}^{3} \alpha_j Z_{|j}$. A way to think about the decomposition is to compare it to a progress bar. In the illustration $Z$ takes value $b_2$, and so the random variables $Z_{|1}$ and $Z_{|2}$ corresponding to the first two segments "light up" (take value 1), whereas the random variable $Z_{|3}$ corresponding to the last segment remains "turned off" (takes value 0). The value of $Z$ equals the sum of the lengths $\alpha_j$ of the "lighted up" segments.

# B  Experimental details

In this section, we provide the details of the datasets in Appendix B.1, our neural network architectures in Appendix B.2, and other details in Appendix B.3. We provide further statistics for all the methods on both datasets in Appendix B.4.

## B.1  Datasets

We perform our evaluation on two datasets, MNIST (LeCun and Cortes, 2010) and Fashion MNIST (Xiao et al., 2017). We will introduce these two datasets in the following.

### B.1.1  MNIST

The MNIST (Modified National Institute of Standards and Technology) dataset is one of the most renowned and widely used datasets in the field of machine learning, particularly for training and testing in the domain of image processing and computer vision. It consists of a large collection of handwritten digit images, spanning the numbers 0 through 9.

The MNIST dataset comprises a total of 70,000 grayscale images of handwritten digits, where the training set has 60,000 images and the test set has 10,000 images. Each image in the dataset is 28x28 pixels, resulting in a total of 784 pixels per image. The images are in grayscale, with pixel values ranging from 0 (black) to 255 (white). Each image is associated with a label from 0 to 9, indicating the digit that the image represents. The images are typically stored in a single flattened array of 784 elements, although they can also be represented in a 28x28 matrix format.

### B.1.2  Fashion MNIST

The Fashion MNIST dataset is a contemporary alternative to the traditional MNIST dataset, created to provide a more challenging benchmark for machine learning algorithms. It consists of images of various clothing items and accessories, offering a more complex and varied dataset for image classification tasks.

The Fashion MNIST dataset contains a total of 70,000 grayscale images, where the training set has 60,000 images and the test set has 10,000 images. Each image in the dataset is 28x28 pixels, resulting in a total of 784 pixels per image. The images are in grayscale, with pixel values ranging from 0 (black) to 255 (white). Each image is associated with one of 10 categories, representing different types of fashion items. The categories are: 1. T-shirt/top 2. Trouser 3. Pullover 4. Dress 5. Coat 6. Sandal 7. Shirt 8. Sneaker 9. Bag 10. Ankle boot. Similar to MNIST, the images are stored in a single flattened array of 784 elements but can also be represented in a 28x28 matrix format.

## B.2  Neural network architectures

For all methods, we adopt a family of factorized Gaussian distributions to model both priors and posteriors, characterized by the form $\pi = \mathcal{N}(w, \sigma\mathbf{I})$ where $w \in \mathbb{R}^d$ denotes the mean vector, and $\sigma$ represents the scalar variance. We use feedforward neural networks for the MNIST dataset (LeCun and Cortes, 2010), while using convolutional neural networks for the Fashion MNIST dataset (Xiao et al., 2017).

Both our feedforward neural network and convolutional neural network are probabilistic, and each layer has a factorized (i.e. mean-field) Gaussian distribution.

Our feedforward neural network has the following architecture:

1. Input layer. Input size: $28 \times 28$ (flattened to 784 features).
2. Probabilistic linear layer 1. Input features: 784, output features: 600, activation: ReLU.
3. Probabilistic linear layer 2. Input features: 600, output features: 600, activation: ReLU.
4. Probabilistic linear layer 3. Input features: 600, output features: 600, activation: ReLU.
5. Probabilistic linear layer 4. Input features: 600, output features: 10, activation: Softmax.

Our convolutional neural network has the following architecture:

1. Input layer. Input size: $1 \times 28 \times 28$.

2. Probabilistic convolutional layer 1. Input channels: 1, output channels: 32, kernel size: 3x3, activation: ReLU.

3. Probabilistic convolutional layer 2. Input channels: 32, output channels: 64, kernel size: 3x3, activation: ReLU.

4. Max pooling layer. Pooling size: 2x2.

5. Flattening layer. Flattens the output from the previous layers into a single vector.

6. Probabilistic linear layer 1. Input features: 9216, output features: 128, activation: ReLU.

7. Probabilistic linear layer 2 (output layer). Input features: 128, output features: 10, activation: Softmax.

## B.3    Other details in the experiments

**General for all methods**    The methods in comparisons are trained and evaluated using the procedure described in Section 2 and visually illustrated in Figure 1. We will provide some further details for each method later in the following. For all methods in comparison, we apply the optimization and evaluation method described in Section 5.1. For the approximation described in Section 5.1.1, we set the parameters $c_1 = c_2 = 5$. The lower bound for the prediction $p_{\min} = 1e - 5$. The $\delta$ in our bound and all the other methods is selected to be $\delta = 0.025$. As mentioned in Section 5.1.2, we use the PAC-Bayes-classic bound by McAllester in replacement of PAC-Bayes-kl when doing optimization. Note that for all methods, we also have to estimate the empirical loss of the posterior $\mathbb{E}_\pi[\cdot]$ described in Section 5.1.3. We also allocate the budget for the union bound for the estimation such that these estimations in the bound are controlled with probability at least $1 - \delta'$, where we chose $\delta' = 0.01$. Therefore, the ultimate bounds for all methods hold with probability at least $1 - \delta - \delta'$. Note that we do not consider such bounds during optimization but only when estimating the bounds.

For all methods, we adopt a family of factorized Gaussian distributions to model both priors and posteriors of all the learnable parameters of the classifiers, characterized by the form $\pi = \mathcal{N}(w, \sigma \mathbf{I})$ where $w \in \mathbb{R}^d$ denotes the mean vector, and $\sigma$ represents the scalar variance. For all methods, we initialize an uninformed prior $\pi_0 = \mathcal{N}(w_0, \sigma_0 \mathbf{I})$ that is independent of data, where the mean is randomly initialized, and the variance $\sigma_0$ is initialized to 0.03 (Pérez-Ortiz et al., 2021).

In the training process of all methods in our experiments, we set the batch size to 250, the number of training epochs to 200, and use stochastic gradient descent with a learning rate of 0.005 and a momentum of 0.95.

**Uninformed priors**    We take $\pi_0$ defined above as the uninformed prior. We then learn the posterior $\rho$ from the prior using the entire training dataset $S$, applying a PAC-Bayes bound. We evaluate the bound using, again, the entire training dataset $S$.

**Data-informed priors**    We start with the same $\pi_0$ as the uninformed prior. We train the informed prior $\pi_1$ using $S_1$ with $|S_1| = |S|/2$ by minimizing a PAC-Bayes bound. The posterior $\rho$ is then learned using the informed prior $\pi_1$ and the subset $S_2$ with $|S_2| = |S|/2$, again by minimizing a PAC-Bayes bound. The bound is evaluated using $S_2$.

**Data-informed priors + excess loss**    We train the informed prior $\pi_1$ and the reference classifier $h^*$ using $S_1$ that contains half of the training dataset. $\pi_1$ is obtained by minimizing a PAC-Bayes bound with the uninformed prior $\pi_0$, while the reference classifier $h^*$ is obtained by an empirical risk minimizer (ERM). The posterior $\rho$ is obtained by minimizing a PAC-Bayes bound on the excess loss between $\rho$ and $h^*$. The prior used in the bound for both training and evaluation is the data-informed prior $\pi_1$. Therefore, the data for both training and evaluation of $\rho$ must be the other half of data $S_2$.

## B.4    Further results for the experiments

In this section, we report some more statistics for all methods.

For all methods, to calculate the classification loss of $\rho$ on the testing data, $\mathbb{E}_\rho[\hat{L}(h, S^{\text{test}})]$ (Test 0-1), we sample one classifier for each data. The train 0-1 loss for all methods is computed on the entire training dataset $S$, while the test 0-1 loss for all methods is computed on the test dataset $S_{\text{test}}$.

### B.4.1 Recursive PAC-Bayes

We report the additional results of Recursive PAC-Bayes on MNIST with $T = 2$ in Table 4, $T = 4$ in Table 5, and $T = 6$ in Table 6. We report Recursive PAC-Bayes on Fashion MNIST with $T = 2$ in Table 7, $T = 4$ in Table 8, and $T = 6$ in Table 9.

Table 4: Insight into the training process of the Recursive PAC-Bayes for $T = 2$ on MNIST.

| $t$ | $n_t^{\text{val}}$ | $\mathbb{E}_{\pi_t}[\hat{F}_{\gamma_t}(h, U_t^{\text{val}} \circ \hat{\pi}_{t-1})]$ | $\frac{\text{KL}(\pi_t^* \| \pi_{t-1}^*)}{n_t^{\text{val}}}$ | $\mathcal{E}_t(\pi_t^*, \gamma_t)$ | $B_t(\pi_t^*)$ | Test 0-1 |
|---|---|---|---|---|---|---|
| 1 | 60000 | | .024 (3e-5) | | .370 (1e-3) | .254 (2e-3) |
| 2 | 30000 | .013 (3e-3) | .024 (1e-4) | .136 (3e-3) | .321 (3e-3) | .139 (3e-3) |

Table 5: Insight into the training process of the Recursive PAC-Bayes for $T = 4$ on MNIST.

| $t$ | $n_t^{\text{val}}$ | $\mathbb{E}_{\pi_t}[\hat{F}_{\gamma_t}(h, U_t^{\text{val}} \circ \hat{\pi}_{t-1})]$ | $\frac{\text{KL}(\pi_t^* \| \pi_{t-1}^*)}{n_t^{\text{val}}}$ | $\mathcal{E}_t(\pi_t^*, \gamma_t)$ | $B_t(\pi_t^*)$ | Test 0-1 |
|---|---|---|---|---|---|---|
| 1 | 60000 | | .023 (8e-5) | | .374 (1e-3) | .258 (1e-3) |
| 2 | 52500 | -4e-4 (1e-3) | .025 (3e-4) | .118 (1e-3) | .305 (1e-3) | .126 (2e-3) |
| 3 | 45000 | .053 (1e-3) | .002 (9e-5) | .087 (2e-3) | .240 (2e-3) | .114 (1e-3) |
| 4 | 30000 | .054 (1e-3) | .001 (2e-5) | .083 (1e-3) | .203 (8e-4) | .109 (1e-3) |

Table 6: Insight into the training process of the Recursive PAC-Bayes for $T = 6$ on MNIST.

| $t$ | $n_t^{\text{val}}$ | $\mathbb{E}_{\pi_t}[\hat{F}_{\gamma_t}(h, U_t^{\text{val}} \circ \hat{\pi}_{t-1})]$ | $\frac{\text{KL}(\pi_t^* \| \pi_{t-1}^*)}{n_t^{\text{val}}}$ | $\mathcal{E}_t(\pi_t^*, \gamma_t)$ | $B_t(\pi_t^*)$ | Test 0-1 |
|---|---|---|---|---|---|---|
| 1 | 60000 | | .019 (7e-5) | | .425 (1e-3) | .311 (3e-3) |
| 2 | 58125 | -0.013 (1e-3) | .032 (6e-4) | .128 (2e-3) | .341 (3e-3) | .139 (1e-3) |
| 3 | 56250 | .050 (1e-3) | .003 (1e-4) | .093 (6e-4) | .264 (1e-3) | .117 (2e-3) |
| 4 | 52500 | .051 (1e-3) | .001 (6e-5) | .080 (9e-4) | .212 (5e-4) | .108 (2e-3) |
| 5 | 45000 | .051 (1e-3) | 9e-4 (3e-5) | .076 (2e-3) | .182 (1e-3) | .104 (6e-4) |
| 6 | 30000 | .049 (1e-3) | 7e-4 (3e-5) | .074 (1e-3) | .166 (1e-3) | .101 (1e-3) |

Table 7: Insight into the training process of the Recursive PAC-Bayes for $T = 2$ on Fashion MNIST.

| $t$ | $n_t^{\text{val}}$ | $\mathbb{E}_{\pi_t}[\hat{F}_{\gamma_t}(h, U_t^{\text{val}} \circ \hat{\pi}_{t-1})]$ | $\frac{\text{KL}(\pi_t^* \| \pi_{t-1}^*)}{n_t^{\text{val}}}$ | $\mathcal{E}_t(\pi_t^*, \gamma_t)$ | $B_t(\pi_t^*)$ | Test 0-1 |
|---|---|---|---|---|---|---|
| 1 | 60000 | | .011 (3e-5) | | .466 (1e-3) | .389 (5e-3) |
| 2 | 30000 | .064 (3e-3) | .013 (2e-4) | .171 (4e-3) | .404 (3e-3) | .266 (5e-3) |

Table 9: Insight into the training process of the Recursive PAC-Bayes for $T = 6$ on Fashion MNIST.

| $t$ | $n_t^{\text{val}}$ | $\mathbb{E}_{\pi_t}[\hat{F}_{\gamma_t}(h, U_t^{\text{val}} \circ \hat{\pi}_{t-1})]$ | $\frac{\text{KL}(\pi_t^* \| \pi_{t-1}^*)}{n_t^{\text{val}}}$ | $\mathcal{E}_t(\pi_t^*, \gamma_t)$ | $B_t(\pi_t^*)$ | Test 0-1 |
|---|---|---|---|---|---|---|
| 1 | 60000 | | 9e-3 (8e-5) | | .534 (4e-3) | .462 (5e-3) |
| 2 | 58125 | .013 (6e-3) | .023 (1e-3) | .151 (4e-3) | .418 (5e-3) | .254 (6e-3) |
| 3 | 56250 | .091 (3e-3) | 3e-3 (4e-4) | .141 (2e-3) | .350 (3e-3) | .223 (1e-3) |
| 4 | 52500 | .090 (2e-3) | 9e-4 (9e-5) | .121 (1e-3) | .296 (1e-3) | .207 (1e-3) |
| 5 | 45000 | .090 (1e-3) | 6e-4 (3e-5) | .117 (2e-3) | .265 (2e-3) | .199 (2e-3) |
| 6 | 30000 | .093 (1e-3) | 5e-4 (2e-5) | .122 (1e-3) | .255 (1e-3) | .198 (1e-3) |

### B.4.2 Uninformed priors

We report the additional results of uninformed priors (McAllester, 1998) on MNIST and Fashion MNIST in Table 10. As described earlier in Section 2, Section 5, and Section B.3, we evaluate the bound using the entire training set.

Table 8: Insight into the training process of the Recursive PAC-Bayes for $T = 4$ on Fashion MNIST.

| $t$ | $n_t^{\text{val}}$ | $\mathbb{E}_{\pi_t}[\hat{F}_{\gamma_t}(h, U_t^{\text{val}} \circ \hat{\pi}_{t-1})]$ | $\frac{\text{KL}(\pi_t^*\|\pi_{t-1}^*)}{n_t^{\text{val}}}$ | $\mathcal{E}_t(\pi_t^*, \gamma_t)$ | $B_t(\pi_t^*)$ | Test 0-1 |
|---|---|---|---|---|---|---|
| 1 | 60000 | | .011 (1e-4) | | .476 (4e-3) | .397 (6e-3) |
| 2 | 52500 | .032 (6e-4) | .017 (9e-4) | .147 (2e-3) | .386 (3e-3) | .240 (4e-3) |
| 3 | 45000 | .100 (3e-3) | 3e-3 (1e-4) | .138 (5e-3) | .331 (4e-3) | .222 (5e-3) |
| 4 | 30000 | .095 (2e-3) | 7e-4 (5e-5) | .128 (2e-3) | .293 (1e-3) | .213 (3e-3) |

Table 10: Further details to compute the bound for the uninformed prior approach on MNIST and Fashion MNIST.

| | $\mathbb{E}_{\rho}[\hat{L}(h, S)]$ | $\frac{\text{KL}(\rho\|\pi_0)}{n}$ | Bound | Test 0-1 |
|---|---|---|---|---|
| MNIST | .343 (2e-3) | .023 (4e-5) | .457 (2e-3) | .335 (3e-3) |
| F-MNIST | .382 (2e-3) | .011 (8e-6) | .464 (2e-3) | .384 (5e-3) |

### B.4.3 Data-informed priors

We report the additional results of data-informed priors (Ambroladze et al., 2007) on MNIST and Fashion MNIST in Table 11. As described earlier in Section 2, Section 5, and Section B.3, we evaluate the bound using $S_2$ that is independent of the data-informed prior $\pi_1$.

Table 11: Further details to compute the bound for the data-informed prior on MNIST and Fashion MNIST.

| | $\mathbb{E}_{\rho}[\hat{L}(h, S_2)]$ | $\frac{\text{KL}(\rho\|\pi_0)}{|S_2|}$ | Bound | Test 0-1 |
|---|---|---|---|---|
| MNIST | .376 (8e-4) | 8e-4 (9e-6) | .408 (9e-4) | .371 (6e-3) |
| F-MNIST | .412 (1e-3) | 4e-4 (7e-6) | .440 (1e-3) | .413 (6e-3) |

### B.4.4 Data-informed priors + excess loss

We report the additional results of data-informed priors + excess loss (Mhammedi et al., 2019) on MNIST and Fashion MNIST in Table 12 and 13. As described earlier in Section 2, Section 5, and Section B.3, we evaluate the bound using $S_2$ that is independent of the data-informed prior $\pi_1$ and the reference prediction rule $h^*$. The bound is composed of two parts: a bound on the excess loss of $\rho$ with respect to $h^*$ (Excess bound) and a single hypothesis bound on $h^*$ ($h^*$ bound). We report the two components of the bound in Table 12. We provide further details to compute these bounds from the losses of their corresponding quantities in Table 13.

Table 12: Details to compute the bound for the data-informed prior and excess loss on MNIST and Fashion MNIST. The table shows the bound on the excess loss of $\rho$ with respect to $h^*$ (Excess bound) and a single hypothesis bound on $h^*$ ($h^*$ bound).

| | Ex. Bound | $h^*$ Bound | Bound | Test 0-1 |
|---|---|---|---|---|
| MNIST | .162 (1e-3) | .029 (4e-4) | .192 (2e-3) | .151 (3e-3) |
| F-MNIST | .196 (5e-3) | .145 (1e-3) | .342 (6e-3) | .285 (5e-3) |

Table 13: Further details to compute the bound for the data-informed prior and excess loss on MNIST and Fashion MNIST. The table shows the empirical excess loss $\mathbb{E}_{\rho}[\hat{\Delta}(h, h^*, S_2)]$, where we define $\Delta(h, h^*, S_2) = \hat{L}(h, S_2) - \hat{L}(h^*, S_2)$, and its bound (Excess Bound). It also shows the empirical loss of the reference prediction rule $\hat{L}(h^*, S_2)$ and its bound. The computation of such bound does not involve the KL term.

| | $\mathbb{E}_{\rho}[\hat{\Delta}(h, h^*, S_2)]$ | $\frac{\text{KL}(\rho\|\pi_1)}{|S_2|}$ | Ex. Bound | $\hat{L}(h^*, S_2)$ | $h^*$ Bound | Bound |
|---|---|---|---|---|---|---|
| MNIST | -0.011 (3e-3) | .035 (5e-4) | .162 (1e-3) | .026 (4e-4) | .029 (4e-4) | .192 (2e-3) |
| F-MNIST | .104 (6e-3) | .018 (5e-4) | .196 (5e-3) | .112 (1e-3) | .145 (1e-3) | .342 (6e-3) |

