# OpenReview forum: "Recursive PAC-Bayes: A Frequentist Approach to Sequential Prior Updates with No Information Loss"
_NeurIPS.cc/2024/Conference — NeurIPS 2024 spotlight_

### Official Review · Reviewer_gLvU · 2024-07-10

**Soundness:** 3
**Presentation:** 3
**Contribution:** 3
**Rating:** 6
**Confidence:** 3

**Summary:**

The paper investigates a type of PAC-Bayes bounds that makes use of sequential prior updates, without loosing the appearance of the cardinality of the whole training set.

**Strengths:**

* Sequential updates feel natural in that context, not only statistically as demonstrated here, but also computationally. Indeed, having sequential Monte Carlo methods in mind, it might be easier to approximate a complex posterior by slowly modifying the prior.
* The first three sections of the paper are very nicely written, and agreeable to read for a non-expert.
* While the proofs are straightforward consequences of earlier results, the key theorem has practical importance and, to my limited knowledge and following the authors' claim, seems to have been missed by earlier investigations.
* Overall, I enjoyed reading the paper, and feel like I learned something.

**Weaknesses:**

## General comments
* The crux of the paper, Section 4, is too dense and hard to read, to the point of hiding some of the degrees of freedom of the paper, and making it tedious to follow the paper line by line and check the claims. This can easily be solved; see major comments.
* After a very nice historical introduction, I was expecting a down-to-earth comparison to recent work on PAC-Bayes and sequential posterior updates, as listed in p2 L60-68. Maybe my lack of expertise made me miss a point, but I think these approaches are introduced and discarded in a single high-level paragraph, which feels unfair compared to the (nice) long pedagogic account of Section 2.

## Major
1. p2 L60 to L68: this paragraph cites several competing martingale-based approaches to PAC-Bayes inequalities with sequential posterior updates, which are discarded a bit quickly for me to understand why. Could you elaborate and give some details on the claims of e.g. Section 6.5 of [Chugg et al., 2023] and [Haddouche and Guedj, 2023]? For instance, you could give an informal version of their main claim, and point to the limitations you address.
2. Relatedly, why are these approaches not featured in the experimental section? Maybe the answer to bullet 1. will make it clear.
3. I would avoid bullet lists on p4 and p7, and put the proofs of Theorems 3 and 4 in the appendix; all of this would create space for Section 4 to breathe. I was enjoying reading the paper until Section 4, and then bumped into the daunting density of Section 4 on p6, due in part to too many inline formulas and overly compact definitions. The risk is to lose many readers here, while this is the crux of the paper. With the space gained, you could e.g. use centered displays for the most important definitions.
4. With the space created, still in Section 4, I would take time to introduce objects one by one. I believe $(\gamma_t)$, for instance, is used before Theorem 5 without definition. Implicitly, we understand in Theorem 5 that they can be predictable functions. We should thus, for instance, have a definition of $\gamma_t$ as a sequence of real-valued functions adapted to a filtration. Then we could introduce, for $\gamma\in\mathbb{R}$, the function $f_{\gamma}: \mathcal{H} \times \mathcal{X} \times \mathcal{Y} \times \mathcal{H} \rightarrow \mathbb{R}$ by $f_{\gamma}(h, x, y, h')=...$, instead of the more compact definition of $f_{\gamma_t}(h, (X,Y,h'))$, which saves space but intertwines the definitions of several objects, functions and random variables. Similarly, I would start by defining a filtration to be able to state which function is measurable w.r.t. to what $\sigma$-algebra.
5. Relatedly, L232 I am not sure I understand over what distribution the expectation is taken. Slower definitions would likely make it clearer.
6. Section 4 feels like it could be rewritten in terms of (super)martingales, although it's a matter of taste, I guess, and I think I saw a comment in the paper that you wanted to avoid doing so. However, not explicitly using martingales makes it harder to connect to previous work. For instance, your theorem 5 bears resemblance with a maximal inequality for martingales such as Ville's, as in [Chugg et al., 2023], could you comment on this?
7. Can you confirm that I understand well Section 5.1? Basically, you describe how you approximate the sequential optimization problems in (4), for $t=1, ..., T$. This does create a sequence $(\pi^*_t)$, to which you can apply Theorem 5. Actually, Theorem 5 does not care whether $(\pi^*_t)$ optimizes (4), so it is OK to actually approximate the optimal sequence (4).
8. Section 5.1.3: how do you draw $h_{t,1}\sim\pi_t^*$? Is $\pi_t^*$ easy to draw from by construction?
9. Section 5: as a side remark, I am wondering whether sequential Monte Carlo techniques could help sequentially approximating $\pi^*_t$; see e.g. [[Chopin and Papaspiliopoulos, An Introduction to Sequential Monte Carlo, Springer, 2020]](https://link.springer.com/book/10.1007/978-3-030-47845-2).
10. p8 L320 what is meant by "Gaussian distributions modeled by probabilistic neural networks"?
11. Why do you not split the data more than in 6 shards? What happens if we go "online" and make $n$ shards of $1$ data point? What differs then from the "online PAC-Bayes" approach of [Haddouche and Guedj, 2023]?
12. Is there any computational gain in splitting the dataset and sequentially optimizing $\pi_t$ instead of doing plain batch PAC-Bayes, on top of the statistical gain in the tighter bound?

## Minor
* p1 L27 The Bayesian posterior

**Questions:**

Any of my major comments above, with a priority to items 1 and 11, and commitment to follow the suggestions of clarification in Section 4.

**Limitations:**

This is fundamental work; no potential negative or environmental societal impact other than making the pdf available!

---

> ### Author Rebuttal · Authors · 2024-08-06
>
> General comments:
> - We agree that Section 4 is too dense. We will take advantage of the tenth page offered for the final version to sparsify it.
> - Concerning works on sequential posterior updates: we are sorry for brevity, let us elaborate. In works on sequential posterior updates the prior remains fixed, and the posterior is updated as the data are processed point-by-point. Our contribution is the sequential update of the prior. It can be directly combined with sequential posterior updates, because the excess loss (the first term in Eq. (2)) can be bounded sequentially by processing the corresponding data chunk point-by-point, exactly as it is done in works on sequential posterior updates. We will add this clarification to the paper.
>
> Major:
> 1. Sequential posterior updates of Chugg et al. (2023) can be combined with our approach, as described above. Concerning their sequential prior updates in Section 6.5, we note that the denominator of the bound reduces to the sample size since the last prior update, meaning that there is no preservation of confidence information, and it is not different from other data-dependent priors. (This may not be clear from the way they have presented the bound, but if you check the proof of the bound, this limitation becomes evident. Specifically, the proof of Thm 4, which is used to prove Thm 40, only allows them to use samples since the last prior update to be able to swap the expectations in the second line on Page 11 in the JMLR version of the paper.) On the other hand, as explained in Lines 64-66, Haddouche and Guedj (2023) provide a generalization bound that applies to an aggregation of posteriors, and their denominator of the generalization bound depends on the number of aggregated posteriors, which can be as large as the number of data points. This is a very different object from a standard posterior studied in our work. In particular, the generalization power of their approach comes primarily from aggregation, and their method requires construction of a large number of posteriors to work. We do not see any close connection between their work and ours. The need to construct and maintain as many posteriors as the number of data points is a limitation in terms of computation and storage. As far as we can see, the authors have not conducted empirical evaluation of their method, presumably because of that.
> 2. Sequential construction of posteriors for every data point is computationally expensive, and experimental evaluations of this approach are highly limited. It would not have been possible to conduct such experiments given the size of the datasets we have worked with and the computational resources available to us.
> 3. We fully agree that Section 4 is too dense. We will use the tenth page given for the final version to implement your suggestions. (We still think that there is value in keeping the proofs in the body, luckily we will not need to compromise on that.)
> 4. Thanks a lot for your valuable suggestions! We will definitely use them to improve the writing.
> 5. Oh, yes. In Line 232 $S$ is sampled according to $\mathcal D$ and $\hat \pi_{t-1}$ is a sequence of prediction rules, one prediction rule per every sample in $S$, sampled according to $\pi_{t-1}$. The expectation is with respect to the two distributions. We will find a better way to present this.
> 6. As already mentioned above, bounding of the excess loss (the first term in Eq. (2)) can be done sequentially using martingale techniques, as in the work of Chugg et al. We will add a comment about this, but we prefer to avoid spelling out the technical details. We believe that for those who are familiar with these techniques, filling out the technical details would be trivial, but for those who are not, it can make the reading unnecessarily complicated and obstruct the main message of our work.
> 7. Yes, this is correct. And, yes, Thm 5 applies to any sequence $\pi_t^*$, so it is not a problem that we only get an approximation of the optimum.
> 8. It depends on the model to which Recursive PAC-Bayes is applied. For our model we borrowed the sampling procedure from Pérez-Ortiz et al., (2021). The details are described in Appendix B.2 and B.3. Since in our case $\pi_t^*$ is a probabilistic neural network represented by a factorized Gaussian distribution, it is easy to sample from.
> 9. That’s a good question. Again, it depends on the models to which Recursive PAC-Bayes is applied.
> 10. To clarify the confusion, replace the sentence in L320 with: “Similar to them we used probabilistic neural networks with weights represented by Gaussian distributions.”
> 11. There are two reasons to limit $T$. The first is that the added value of additional splits saturates (the improvement in going from $T=4$ to $T=6$ is already relatively small), so at some point the cost of the union bound on the number of splits (the $\ln T$ factor in the bound) will dominate the benefit of making extra splits. The second reason is computational. Once the improvements saturate, it makes little sense to split further. We will add experiments with $T=8$, which show that there is little gain relative to $T=6$.
> 12. The computational cost or gain depends on the models to which the bound is applied in two ways. First, if the cost of finding the (approximate) argmin in Eq. (4) is linear in the sample size, as in our experiments, then the difference between using or not using the recursion is small. If the method would have been applied to models, where the cost of finding the argmin is superlinear (e.g., kernel SVMs), then it could lead to substantial computational savings. The second source for potential computational gain is a trade-off with the statistical gain. Specifically, we could imagine a scenario, where compromising on the statistical gain and doing several recursion steps with more relaxed approximation of the argmin could still yield better and faster outcomes than doing a more precise single-step posterior optimization.

---

> > ### Comment · Reviewer_gLvU · 2024-08-10
> >
> > Thanks for the clarifications!

---

### Official Review · Reviewer_ZaLx · 2024-07-10

**Soundness:** 3
**Presentation:** 3
**Contribution:** 3
**Rating:** 7
**Confidence:** 3

**Summary:**

Data-dependent priors are crucial for the tightness of PAC-Bayesian bounds in several scenarios. However, using a fraction of the data to train the prior reduces the sample size of the bounds, which can sometimes be counter-productive. This fact also discourages sequential prior updates since the bounds can rapidly degenerate. This paper elegantly solves these problems using a "prior-dependent" excess loss bound inspired by Mhammedi et al. (2019). Their approach allows sequential prior updating without confidence loss and surpasses previous data-dependent prior building techniques in standard benchmark datasets.

Additionally, they generalize ternary split-kl PAC-Bayes bounds to general discrete random variables.

**Strengths:**

1- The main result, Theorem 5, is a significant contribution to the study of data-dependent priors with many possible applications.

2- The experimental results (Table 1) show a remarkable improvement over previous methods.

**Weaknesses:**

I find no major weaknesses, just a couple of comments:

1- Sections 3 and 4 are very dense in notation and difficult to read (I recognize there is not much the authors can do about that).

2- Since Recursive PAC-Bayes is developed for sequential prior updating, it would be nice (if possible) to have some discussion about its possible applications to streaming/online learning scenarios.

3- In the experiments with data-dependent priors 1/2 of the data was used for building the prior. However [1] shows that the proportion of data used for prior building is crucial. It would be nice to have experiments with different data splitting proportions (also, I think [1] is a relevant paper that could be cited as related work).



[1] Dziugaite, G. K., Hsu, K., Gharbieh, W., Arpino, G., & Roy, D. (2021). On the role of data in PAC-Bayes bounds. In International Conference on Artificial Intelligence and Statistics (pp. 604-612). PMLR.

**Questions:**

See Weaknesses.

**Limitations:**

No significant limitations.

---

> ### Author Rebuttal · Authors · 2024-08-06
>
> We thank the reviewer for their feedback.
>
> 1. We agree that Section 3 and, even more crucially, Section 4 are too dense. We will take advantage of the tenth page offered for the final version to sparcify writing.
> 2. The application to streaming and online learning is essentially straightforward, because what we do with the data is effectively sequential processing. We will add a comment to the paper.
> 3. We fully agree that the splitting proportions matter. Splitting half-half is sort of default, and our approach can be seen as a default generalization of it, because we recursively continue splitting the first half further. Of course, experimenting with other proportions would be a natural continuation, although we think that it might be more appropriate for a journal extension, both due to space constraints, and because explosion in the number of experiments risks distructing from the primary theoretical contribution of the paper.

---

> > ### Comment · Reviewer_ZaLx · 2024-08-11
> >
> > Thank you for your response, I am happy with the feedback and I will maintain my score. My only suggestion would be to use the extra page in the camera-ready version to extend Sections 3 and 4 to make them more readable. I wish the authors the best.

---

### Official Review · Reviewer_Kzqj · 2024-07-10

**Soundness:** 3
**Presentation:** 2
**Contribution:** 3
**Rating:** 6
**Confidence:** 4

**Summary:**

- The paper presents a novel PAC-Bayesian procedure that allows for sequential prior updates without splitting the training dataset.
- The proposed procedure is based on a novel decomposition of the expected loss of randomized classifiers, which rewrites the posterior loss as an excess loss relative to a downscaled loss of the prior plus the downscaled loss of the prior, which is recursively bounded.
- The paper generalizes the split-kl and PAC-Bayes-split-kl inequalities to discrete random variables.
- The authors confirmed that their PAC-Bayes learning method outperforms state-of-the-art methods in empirical evaluations.

**Strengths:**

- The paper addresses a practically important topic of adaptively optimizing the prior in PAC-Bayes learning to reduce pre-training costs while enhancing generalization performance.
- The attempt to derive recursively bounded generalization error bounds for the optimization steps includes originality.

**Weaknesses:**

First, I want to note that I find it difficult to fully grasp the motivation, utility, and specific algorithms of this paper from the current manuscript, which may lead to some of the points in the Weaknesses and Questions sections being based on misunderstandings. If there are any misunderstandings, I would appreciate it if the authors could correct them and provide clearer explanations beyond what is presented in the paper.

- **Limitations of Derived Bounds**:
  - As one of the limitations in this paper, the derived bounds assume 0-1 loss, limiting their applicability to binary classification problems.
  - The bounds are valid only at a specific time point $t$. Considering that most existing PAC-Bayes bounds provide insights into generalization error independent of iteration, this bound appears to be valid under very restrictive conditions.
  - The definitions of symbols throughout the paper seem ambiguous, affecting readability. For instance, it is unclear what $t$ represents—whether it is training iteration, epoch, or a separate recursion step. Clarification is needed.

- **Comparison with Existing Approaches**:
  - The main issue addressed by this paper is that traditional data-dependent priors in PAC-Bayes learning are constructed by splitting the training data, thus discarding some information. However, alternative approaches not discussed in this paper use optimization methods like SGLD or Entropy-SG(L)D, which guarantee differential privacy while optimizing the prior using all training data (e.g., Dziugaite et al., 2017; Dziugaite et al., 2018). Although these methods might result in looser bounds or smaller probabilities of bound validity, they have shown good performance in similar experimental settings. While these methods differ in being pre-training approaches rather than recursive, they share the goal of optimizing the prior without splitting the training data. The lack of comparison with these methods leaves the usefulness of the proposed approach insufficiently demonstrated.

- **Data Splitting and Recursive Approach**:
  - The paper still splits data according to the value of $T$. It is unclear how this fundamentally differs from traditional data-splitting prior learning approaches. In the split strategy proposed in Sec. 5, increasing the number of split data sets with $T$ might lead to suboptimal prior distributions due to fewer data in early stages, potentially affecting generalization performance.
  - There is no discussion on the tightness of the bounds derived in Theorem 5, especially for $t \geq 2$. Intuitively, even though $\gamma_{t}$ is multiplicative, the cumulative addition of the KL term may result in looser bounds as $t$ increases unless $\gamma_{t}$ or/and the complexity value is very small. The decreasing bound values with increasing $T$ observed in the experiments may be due to the simplicity of datasets like MNIST, where KL values are initially very small. If there is another reason, it should be explained.

- **Computational Efficiency**:
  - There is no mention of computational efficiency. It seems that increasing the number of recursive steps would lead to higher computational costs.

- **Convergence of generalization bounds**:
  - It is unclear whether the bound guarantees convergence with respect to the sample size $n$, i.e., whether it converges to zero as $n \rightarrow \infty$. If we only consider Equation (3), even if $n_{t}^{\mathrm{val}} \rightarrow \infty$ and $B_{t-1} = 0$, it seems that $-\gamma_{t}$ would remain in $\mathcal{E}_{t}$, preventing convergence. Is this a desirable property for a generalization error bound?

- **Experimental Validation**:
  - The experiments are limited to very simple datasets like MNIST and Fashion-MNIST, which is insufficient to convincingly demonstrate the utility of the proposed algorithm. More diverse and complex datasets should be tested to validate the approach comprehensively.

Citation:
- Dziugaite et al., 2017: Dziugaite et al. Data-dependent PAC-Bayes priors via differential privacy. NeurIPS2018. https://arxiv.org/pdf/1802.09583
- Dziugaite et al., 2018: Dziugaite et al. Entropy-SGD optimizes the prior of a PAC-Bayes bound: Generalization properties of Entropy-SGD and data-dependent priors. ICML2018. https://arxiv.org/pdf/1712.09376

**Questions:**

In addition to the concerns raised in the Weakness section, I would appreciate it if you could also address the following questions:

- Can you provide a qualitative interpretation of the generalization error bound given by Equation (3)?
- Why does the summation part in \(\mathcal{E}\) above Equation (3) only sum from \(j=1\) to \(j=3\)?
- The specific algorithm of the proposed method is not very clear. Could you provide an algorithm table?
- What are the advantages of the proposed method compared to traditional PAC-Bayes prior pre-training approaches that fully use the training data (e.g., Dziugaite et al., 2017; Dziugaite et al., 2018)? If there is a reason for not conducting a theoretical or experimental comparison with these methods, please explain.
- You mentioned calculating the 0-1 loss. Is this correct that you conducted binary classification using a one-vs-all approach?

**Limitations:**

This study is foundational research aimed at improving the generalization by PAC-Bayes learning with prior optimization, and these aspects are discussed in the Broader Impact paragraph. The datasets used are open datasets such as MNIST and Fashion MNIST, which suggests that there are no significant concerns regarding potential negative social impacts.

---

> ### Author Rebuttal · Authors · 2024-08-06
>
> There are numerous and significant misunderstandings in the review, which we clarify below one-by-one.
>
> The first sentence of the Summary states: “The paper presents a novel PAC-Bayesian procedure that allows for sequential prior updates without splitting the training dataset.” This statement is incorrect. We do split the dataset into $T$ folds. The novelty of our work is it allows meaningful sequential processing of the folds without losing confidence information along the way. Prior work could only meaningfully process splitting in two folds, but even then it was losing confidence information.
>
> Limitations of derived bounds:
> - The derived bounds assume zero-one loss: this statement is not fully correct. The main contribution of our paper is the recursive decomposition of the loss in Eq. (2), $\mathbb E_{\pi_t}[L(h)] = \mathbb E_{\pi_t}[L(h) - \gamma_t \mathbb E_{\pi_{t-1}}[L(h)]] + \gamma_t \mathbb E_{\pi_{t-1}}[L(h)]$. It applies to any loss function (as long as the expectations are well-defined). Any suitable PAC-Bayesian bounds can be used to bound the two terms in the decomposition, as mentioned in Lines 167-169. We have restricted the presentation to the zero-one loss and used the most basic PAC-Bayes-kl bound, because otherwise the reader would get lost in variations of PAC-Bayes and miss the main message. But it is straightforward to use, say, PAC-Bayes-Empirical-Bernstein, and obtain a bound for any bounded loss function.
> - The bounds are valid only at a specific time point $t$: First, we believe that there is a confusion between $t$, which indexes folds and recursion steps, and $i$, which indexes samples. Our bounds hold for all $t$, because we take a union bound (factor $T$ under the logarithm in Thm 5). Concerning bounds that hold for all $i$, it is straightforward to replace our bounds for the two terms in the decomposition in Eq. (2) with existing PAC-Bayes bounds that hold for all $i$.
> what $t$ represents: $t$ indexes the folds, which also correspond to recursion steps, because each recursion step processes a new fold.
>
> Comparison with existing approaches: priors based on differential privacy, as in the work of Dziugaite et al., pursue a very different goal from our work. Their goal is to construct a set of interesting priors based on a fixed dataset without revealing too much information. Our goal is to refine the priors while moving from one chunk of data to the next, without losing confidence information.
>
> Data splitting and recursive approach:
> - It is unclear how this fundamentally differs from traditional data-splitting prior learning approaches: Traditional approaches split the data in two folds. Our approach supports meaningful splitting into an arbitrary number of folds.
> - Increasing the number of split data sets with $T$ might lead to suboptimal prior distributions due to fewer data in early stages, potentially affecting generalization performance: Allocating little data to early stages is actually an advantage, as can also be seen from Tables 1, 2, and 3 in the paper (compare small values of $T$ with large values of $T$). This way few data are used to steer the prior into the right region, and then there is still a lot of data to obtain a tight bound.
> - The decreasing bound values with increasing $T$: As you can see from Tables 2 and 3, the first rounds of recursion are used to steer the prior into the right region. The KL term in the first rounds is large, but with every subsequent round the contribution of this term to the final bound is decreased by a factor of $\gamma_t$. Therefore, the larger is the number of rounds, the smaller is the contribution of the first terms to the final bound. In the later rounds the prior is already good, and the KL term is very small (Tables 2 and 3).
>
> Computational Efficiency: See our reply to Reviewer gLvU, Point 12.
>
> Convergence: The decomposition of the excess loss into a superposition of binary variables is exact (see the illustration in Fig. 2 in the appendix). Therefore, if the excess loss converges to zero and $n\to\infty$, then the bound will converge to zero. Put attention that as $n\to\infty$ the $kl^{-1,+}$ terms in the bound converge to the value of the first argument, and not to zero. So, if the excess loss is zero, they will counterbalance $-\gamma_t$.
>
> Experimental validation: we emphasize that the primary contribution of the paper is theoretical, and that it is a conference submission. Further experimental evaluation would be natural for a journal extension.
>
> Questions:
> - Interpretation of the generalization bound in Eq. (3): look at the decomposition of the loss in Eq. (2). The first term in Eq. (3) is a PAC-Bayes bound on the first term in Eq. (2), and the second is a recursive PAC-Bayes bound on the second term in Eq. (2).
> - Why the summation above Eq. (3) goes from $j=1$ to $j=3$: The excess loss takes 4 values ($K=4$, see Line 223) and the corresponding binary decomposition is given in Line 196. You can also check the illustration in Fig. 2 in the appendix.
> - Algorithm: Our algorithm is a recursive computation of the argmin in Eq. (4). The computation of the argmin depends on the model to which the bound is applied, which is external to the paper.
> - Comparison to Dziugaite et al.: addressed above.
> - Clarification concerning the experiments: the experiments apply the zero-one loss in multiclass classification. (Correct prediction gives loss zero, incorrect one.) It is NOT one-vs-all.
>
> Finally, we would like to say that the score given by the reviewer “3: Reject: For instance, a paper with technical flaws, weak evaluation, inadequate reproducibility and/or incompletely addressed ethical considerations.” is unjustified. The reviewer has not identified any technical flaws; our evaluation is reasonable for a theoretical contribution to a conference; we have provided the code to reproduce the experiments and no reproducibility issues were raised; and there are no ethical considerations concerning our work.

---

> > ### Comment · Reviewer_Kzqj · 2024-08-11
> > **Acknowledgements and Apologies**
> >
> > First of all, I deeply apologize for submitting a low-quality review due to my misreading. After reading your response, I have gained a clearer understanding of the content. Thank you for the detailed explanation. As a result, I have decided to change my score to a 6.
> >
> > However, I still have some concerns regarding the authors’ claim that the paper focuses on theoretical contributions because the title explicitly mentions handling "sequential prior updates", which sounds proposing some novel theoretically-validated algorithm. In the abstract, it states, “We present a novel and, in retrospect, surprisingly simple and powerful PAC Bayesian procedure that allows sequential prior updates…” Upon first reading this, I had the impression that the paper would derive an algorithm with a theoretical background and empirically validate it.
> > With this premise, I proceeded with my review, which led me to feel that the paper lacked a numerical comparison based on sequential prior updates and pre-training-based prior distribution settings proposed by some existing studies. This perceived “weakness of evaluation” led me to assign a score of 3.
> >
> > I now realize that my understanding was flawed, and I agree that the score was too low given the paper’s contributions. I apologize for any discomfort this may have caused. If I may suggest one revision, it would be to more clearly present that the primary objective is the theoretical contribution. This could help reduce the risk of misinterpretation by readers like myself and increase the paper’s impact.
> >
> > I wish the authors the best of luck.

---

> > > ### Author Response · Authors · 2024-08-12
> > >
> > > We thank the reviewer for raising the score. We will use input from all the reviewers to improve the presentation.

---

### Official Review · Reviewer_oqWC · 2024-07-13

**Soundness:** 4
**Presentation:** 4
**Contribution:** 4
**Rating:** 8
**Confidence:** 4

**Summary:**

PAC-Bayes bounds are extended to the recursive or streaming case by breaking the expected loss into ($a$) expected loss on the previous step times a discounting factor plus ($b$) current expected loss minus $a$. An extension of the kl inequality to many-valued (here 4-valued) random variables provides a bound on $b$.

**Strengths:**

This is impressive work. It's technically sound and makes a major theoretical advance.

**Weaknesses:**

The proof of thm 5 doesn't account for sampling error between $\hat{\pi}^*_{t-1}$ and $\pi^*_{t-1}$. The inequality used in line 244 refers only to $\pi^*_{t-1}$ but the definitions of
$\mathcal{E}\_{t}$
and
$\\hat{F}\_{\\gamma_{t} | j}$
involve
$\hat{\pi}^{*}\_{t-1}$.
The method in section 5.1.3 can resolve this but it should be included in the theorem (or the theorem should be rewritten without sampling).

214: "can depend on $S$" is conceptually helpful but technically superfluous. What's really meant is that because the quantification is over all of $\mathcal{P}$ one is free to choose $\rho$ in a way that depends on $S$.

231: the summation index is misleading because it implies the index set is the product of $U^{val}\_t$
and $\hat{\pi}\_{t-1}$,
whereas the actual index is $((X,Y),h')$.

**Questions:**

The objective in (4) is very similar to the VI objective, especially with 0-1 loss replaced by cross-entropy as in section 5.1.1. (I think replacing $kl^{-1,+}$ with Euclidean divergence and $n^{val}_t$ with $|S_t|$ would yield the VI update.) It would be useful to see an experimental comparison between recursive PAC-Bayes and recursive VI.

**Limitations:**

none noted

---

> ### Author Rebuttal · Authors · 2024-08-06
>
> Thank you for the detailed review and the feedback.
>
> “sampling error between $\hat \pi_{t-1}^*$ and $\pi_{t-1}^*$” - It is true that this is a delicate point, but in fact no correction is required. We have explained it in lines 219-233, but we accept that the explanation might have been too dense. Let us explain it again. $\mathcal{E}\_t(\pi_t,\gamma_t)$ in Theorem 5 is a summation of PAC-Bayes-kl bounds on $\mathbb{E}\_{\pi_t}[F\_{\gamma_t|j}(h,\pi\_{t-1}^*)]$. By definition, $\hat F\_{\gamma\_t|j}(h,U\_t^{val},\hat \pi\_{t-1}^*)$, which involves the sampling according to $\pi_{t-1}^*$, is an unbiased sample of size $n_t^{val}$ from $F_{\gamma_t|j}(h,\pi_{t-1}^*)$. In contrast to standard applications of PAC-Bayes, where the samples are pairs $(X,Y)$, here the samples are triplets $(X,Y,h’)$, where $h’$ is sampled according to $\pi_{t-1}^*$, but this does not matter for the bound. We will emphasize this in the final manuscript.
>
> Lines 214 & 231: thanks for the comments, we will polish the writing.
>
> The definition of Recursive VI and comparison to Recursive PAC-Bayes could be an interesting direction for future work. We note, though, that while the two share the similarity of using KL regularization, the settings and objectives are quite different. PAC-Bayes aims to minimise the generalization error, whereas VI aims to compute the Bayesian posterior. Therefore, it is not immediately clear what could be a meaningful comparison.

---

### Decision · Program_Chairs · 2024-09-25

**Decision:**

Accept (spotlight)

**Comment:**

The reviewers agreed this is a technically solid paper and a significant advance for the PAC-Bayes learning theory. There is no doubt that it deserves acceptance.

It is also worth mentioning that there is a consensus that the submitted manuscript contains dense explanations, and some parts deserve to be clarified. For the benefit of the readers, I strongly recommend the authors address these concerns in their camera-ready.